# The Temporal Structure of Language Processing in the Human Brain Corresponds to The Layered Hierarchy of Deep Language Models

## Abstract

Deep Language Models (DLMs) provide a novel computational paradigm for understanding the mechanisms of natural language processing in the human brain. Unlike traditional psycholinguistic models, DLMs use layered sequences of continuous numerical vectors to represent words and context, allowing a plethora of emerging applications such as human-like text generation. In this paper we show evidence that the layered hierarchy of DLMs may be used to model the temporal dynamics of language comprehension in the brain by demonstrating a strong correlation between DLM layer depth and the time at which layers are most predictive of the human brain. Our ability to temporally resolve individual layers benefits from our use of electrocorticography (ECoG) data, which has a much higher temporal resolution than noninvasive methods like fMRI. Using ECoG, we record neural activity from participants listening to a 30-minute narrative while also feeding the same narrative to a high-performing DLM (GPT2-XL). We then extract contextual embeddings from the different layers of the DLM and use linear encoding models to predict neural activity. We first focus on the Inferior Frontal Gyrus (IFG, or Broca's area) and then extend our model to track the increasing temporal receptive window along the linguistic processing hierarchy from auditory to syntactic and semantic areas. Our results reveal a connection between human language processing and DLMs, with the DLM's layer-by-layer accumulation of contextual information mirroring the timing of neural activity in high-order language areas.

## 1 Introduction

Autoregressive Deep language models (DLMs) provide an alternative computational framework for how the human brain processes natural language (Goldstein et al., 2022; Caucheteux & King, 2022; Schrimpf et al., 2021; Yang et al., 2019a). Classical psycholinguistic models rely on rule-based manipulation of symbolic representations embedded in hierarchical tree structures (Lees, 1957; Kako & Wagner, 2001). In sharp contrast, autoregressive DLMs encode words and their context as continuous numerical vectors—i.e., embeddings. These embeddings are constructed via a sequence of nonlinear transformations across layers, to yield the sophisticated representations of linguistic structures needed to produce language (Radford et al., 2019; Brown et al., 2020; Yang et al., 2019b; Kulshreshtha et al., 2020; Antonello et al., 2021). These layered models and their resulting representations enable a plethora of emerging applications such as language translation and human-like text generation.

Autoregressive DLMs embody three fundamental principles for language processing: (1) embedding-based contextual representation of words; (2) next-word prediction; and (3) error correction-based learning. Recent research has begun identifying neural correlates of these computational principles in the human brain as it processes natural language. First, contextual embeddings derived from DLMs provide a powerful model for predicting the neural response during natural language processing (Goldstein et al., 2022; Schrimpf et al., 2021; Caucheteux et al., 2023). For example, brain embeddings recorded in the Inferior Frontal Gyrus (IFG) seem to align with contextual

embeddings derived from DLMs (Goldstein et al., 2022; Caucheteux et al., 2023). Second, spontaneous pre-word-onset next-word predictions were found in the human brain using electrophysiology and imaging during free speech comprehension (Goldstein et al., 2022; Donhauser & Baillet, 2020; Heilbron et al., 2022). Third, an increase in post-word-onset neural activity for unpredictable words has been reported in language areas (Goldstein et al., 2022; Weissbart et al., 2020; Willems et al., 2015). Likewise, for unpredictable words, linear encoding models trained to predict brain activity from word embeddings have also shown higher performance hundreds of milliseconds after word onset in higher order language areas (Goldstein et al., 2022), suggesting an error response during human language comprehension. These results highlight the potential for autoregressive DLMs as cognitive models of human language comprehension (Goldstein et al., 2022; Caucheteux & King, 2022; Schrimpf et al., 2021).

Our current study provides further evidence of the shared computational principles between autoregressive DLMs and the human brain, by demonstrating a connection between the internal sequence of computations in DLMs and the human brain during natural language processing. We explore the progression of nonlinear transformations of word embeddings through the layers of deep language models and investigate how these transformations correspond to the hierarchical processing of natural language in the human brain.

Recent work in natural language processing (NLP), has identified certain trends in the properties of embeddings across layers in DLMs (Manning et al., 2020; Rogers et al., 2020; Tenney et al., 2019). Embeddings at early layers most closely resemble the static, non-contextual input embeddings (Ethayarajh, 2019) and best retain the original word order (Liu et al., 2019). As layer depth goes up, embeddings become progressively more context-specific and sensitive to long-range linguistic dependencies among words (Tenney et al., 2019; Cui & Zhang, 2019). In the final layers, embeddings are typically specialized for the training objective (next-word prediction in the case of GPT2–3) (Radford et al., 2019; Brown et al., 2020). These properties of the embeddings emerge from a combination of the architectural specifications of the network, the predictive objective, and the statistical structure of real-world language (Goldstein et al., 2022; Hasson et al., 2020).

In this study, we investigate how the layered structure of DLM embeddings maps onto the temporal dynamics of neural activity in language areas during natural language comprehension. Naively, we may expect the layer-wise embeddings to roughly map onto a cortical hierarchy of language processing (similar to the mapping observed between convolutional neural networks and the primate ventral visual pathway (Guclu & van Gerven, 2015; Yamins & DiCarlo, 2016)). In such a mapping, early language areas would be better modeled by embeddings extracted from early layers of DLMs, whereas higher-order areas would be better modeled by embeddings extracted from later layers of DLMs.

Interestingly, fMRI studies examining the layer-by-layer match between DLM embeddings and brain activity have observed that intermediate layers provide the best fit across many language Regions of Interest (ROIs) (Schrimpf et al., 2021; Toneva & Wehbe, 2019; Caucheteux & King, 2022; Kumar et al., 2022). These findings do not support the hypothesis that DLMs capture the sequential, word-by-word processing of natural language in the human brain.

In contrast, our work leverages the superior spatiotemporal resolution of electrocorticography (ECoG, (Yi et al., 2019; Sahin et al., 2009)), to show that the human brain's internal temporal processing of a spoken narrative matches the internal sequence of nonlinear layer-wise transformations in DLMs. Specifically, in our study we used ECoG to record neural activity in language areas along the superior temporal gyrus and inferior frontal gyrus (IFG) while human participants listened to a 30-minute spoken narrative. We supplied this same narrative to a high-performing DLM (GPT2-XL, (Radford et al., 2019; Brown et al., 2020)) and extracted the contextual embeddings for each word in the story across all 48 layers of the model. We compared the internal sequence of embeddings across the layers of GPT2-XL for each word to the sequence of word-aligned neural responses recorded via ECoG in the human participants. To perform this comparison, we measured the performance of linear encoding models trained to predict temporally-evolving neural activity from the embeddings at each layer. Our performance metric is the correlation between the true neural signal across words at a given lag, and the neural signal predicted by our encoding models under the same conditions.

In our experiments, we first replicate the finding that intermediate layers best predict cortical activity (Schwartz et al., 2019; Caucheteux et al., 2021). However, the improved temporal resolution of

our ECoG recordings reveals a remarkable alignment between the layer-wise DLM embedding sequence and the temporal dynamics of cortical activity during natural language comprehension. For example, within the inferior frontal gyrus (IFG) we observe a temporal sequence in our encoding results where earlier layers yield peak encoding performance earlier in time relative to word onset, and later layers yield peak encoding performance later in time. This finding suggests that the transformation sequence across layers in DLMs maps onto a temporal accumulation of information in high-level language areas. In other words, we find that the spatial, layered hierarchy of DLMs may be used to model the temporal dynamics of language comprehension. We apply this model to other language areas along the linguistic processing hierarchy and validate existing work, which suggests an accumulation of information over increasing time scales, moving up the hierarchy from auditory to syntactic and semantic areas (Hasson et al., 2008).

Our findings show a strong connection between the way the human brain and DLMs process natural language. There are still crucial differences, perhaps the most significant being that the hierarchical structure of DLMs is spatial in nature, whereas that of the brain is spatiotemporal. However, in the discussion we outline potential augmentations of existing DLM architectures that could lead to the next generation of cognitive models of language comprehension.

**Main contributions**    Here we highlight two main contributions of this paper.

1. We provide the first evidence (to the best of our knowledge) that the layered hierarchy of DLMs like GPT2-XL can be used to model the temporal hierarchy of language comprehension in a high order human language area (Broca's Area: Fig. 2). This suggests that the computations done by the brain over time during language comprehension can be modeled by the layer-wise progression of the computations done in the DLM.

2. We further validate our model by applying it to other language related brain areas (Fig. 3). These analyses replicate neuroscientific results that suggest an accumulation of information along the linguistic processing hierarchy and therefore show GPT2-XL's promise not only as a model of the temporal hierarchy within human language areas, but also of the spatial hierarchy of these areas in the human brain.

## 2    PRIOR WORK

Prior studies reported shared computational principles (e.g., prediction in context and representation using a multidimensional embedding space) between DLMs and the human brain (Goldstein et al., 2022; Caucheteux & King, 2022; Schrimpf et al., 2021; Tikochinski et al., 2023). In the current study, we extracted the contextual embeddings for each word in a chosen narrative across *all* 48 layers and fitted them to the neural responses to each word.

A large body of prior work has implicated the IFG in several aspects of syntactic processing (Hagoort, 2005; Grodzinsky & Santi, 2008; Friederici, 2011) and as a core part of a larger-scale language network (Hagoort & Indefrey, 2014; Schell et al., 2017). Recent work suggests that these syntactic processes are closely intertwined with contextual meaning (Schrimpf et al., 2021; Matchin & Hickok, 2019). We interpret our findings as building on this framework: the lag-layer correlations we observe reflect, in part, the increasing contextualization of the meaning of a given word (which incorporates both syntactic and contextual semantic relations) in the IFG, rapidly over time. This interpretation is also supported by recent computational work that maps linguistic operations onto deep language models (Rogers et al., 2020; Manning et al., 2020).

Prior work indicates that the ability to encode the neural responses in language areas using DLMs varies with the accuracy of their next-word predictions and is lower for incorrect predictions (Goldstein et al., 2022; Caucheteux & King, 2022). In contrast, we observe that even for unpredictable words, the temporal encoding sequence was maintained in high-order language areas (IFG and TP) 4. However, we do find a difference in the neural responses for unpredictable words in the IFG. More specifically, for unpredictable words, the time of peak encoding performance for early layers occurs hundreds of milliseconds after onset, whereas for predictable words it occurs just after onset. This may be evidence for later, additional processing of unpredictable words in the human brain as part of an error-correcting mechanism.

Intermediate layers are thought to capture best the syntactic and semantic structure of the input (Hewitt & Manning, 2019; Jawahar et al., 2019) and generally provide the best generalization to other NLP tasks (Liu et al., 2019). Strengthening the findings of prior studies (Schrimpf et al., 2021; Caucheteux et al., 2021; Schwartz et al., 2019), we also noticed that intermediate layers best matched neural activity in language areas. This superior correlation between neural activity and DLMs intermediate layers suggests that the language areas place additional weight on such intermediate representations. At the same time, each layer's embedding is distinct and represents different linguistic dimensions (Rogers et al., 2020), thus invoking a unique temporal encoding pattern (that is also hinted at in Caucheteux & King (2022)). Overall, our finding of a gradual sequence of transitions in language areas is complementary and orthogonal to the encoding performance across layers.

## 3 EXPERIMENTAL DESIGN

### 3.1 NEURAL DATA AND GPT2-XL EMBEDDINGS

We collected electrocorticography (ECoG) data from nine epilepsy patients while they listened to a 30-minute audio podcast ("Monkey in the Middle", NPR 2017). In prior work, embeddings were taken from the final hidden layer of GPT2-XL to predict brain activity and it was found that these contextual embeddings outperform static (i.e. non-contextual) embeddings (Goldstein et al., 2022; Schrimpf et al., 2021; Caucheteux et al., 2021). In this paper, we expand upon this type of analysis by modeling the neural responses for each word in the podcast using contextual embeddings extracted from each of the 48 layers in GPT2-XL (Fig. 1A).

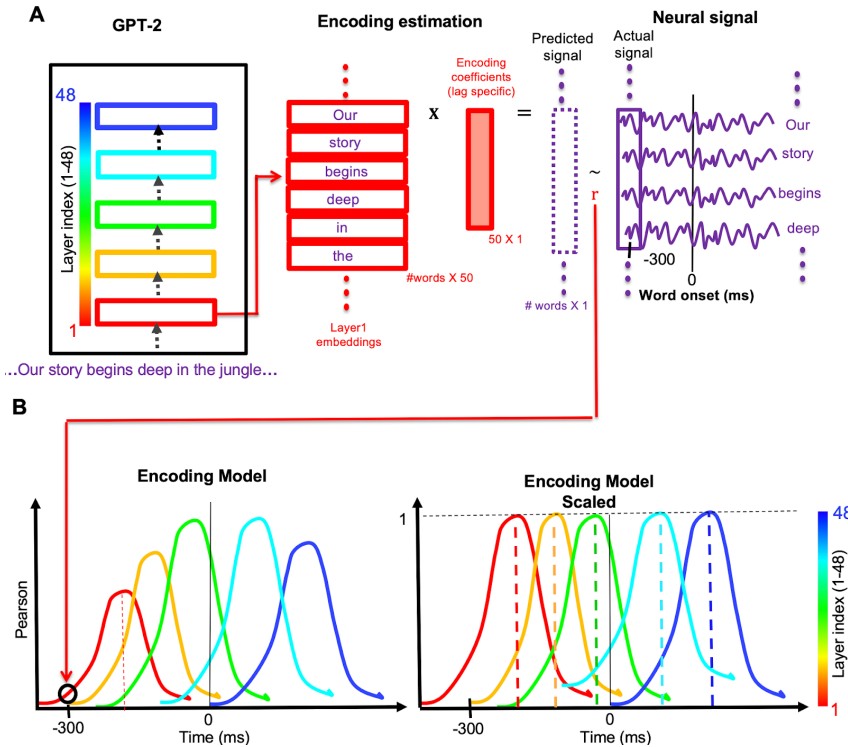

Figure 1: Layer-wise encoding models. (A) Estimating encoding models for each combination of electrode, lag, and layer: A linear model is trained to estimate the lagged brain signal at an electrode from the word embedding at a designated layer. (B, left) Illustration of encoding performance plot. (B, right) Illustration of scaled encoding performance plot.

We focused on four areas along the ventral language processing stream (Hickok & Poeppel, 2004; Karnath, 2001; Poliva, 2016): middle Superior Temporal Gyrus (mSTG) with 28 electrodes, anterior

Superior Temporal Gyrus (aSTG) with 13, Inferior Frontal Gyrus (IFG) with 46, and the Temporal Pole (TP) with 6. We selected electrodes that had significant encoding performance for static embeddings (GloVe) (corrected for multiple comparisons). Finally, given that prior studies have reported improved encoding results for words correctly predicted by DLMs (Goldstein et al., 2022; Caucheteux & King, 2022), we separately modeled the neural responses for predictable words and unpredictable words. We considered words to be predictable if the correct word was assigned the highest probability by GPT2-XL. There are 1709 of these words in the podcast, which we refer to as the top-1 predictable, or just predictable, words. To further separate predictable and unpredictable words and to match the statistical power across the two analyses, we defined unpredictable words as cases where all top-5 next-word predictions were incorrect. In other words, the correct word was not in the set of 5 most probable next words as determined by GPT2-XL. There are 1808 of these words, which we refer to as top-5 unpredictable, or just unpredictable, words. We show results from running our analysis for unpredictable words in Supp Fig. 4. In Supp Figs. 5-7 we show our encoding results for predictable, unpredictable and all words.

## 3.2 ENCODING MODEL

For each electrode, we extracted 4000 ms windows of neural signal around each words' onset (denoted lag 0). The neural signal was averaged over a 200 ms rolling window with incremental shifts of 25 ms. The words and their corresponding neural signals were split into 10 non-overlapping subsets, for a 10-fold cross-validation training and testing procedure. For each word in the story, a contextual embedding was extracted from the output of each layer of GPT-2 (for example, layer 1: red). The dimension of the embeddings was reduced to 50 using PCA. We performed PCA per-layer to avoid mixing information between the layers. We describe the PCA procedure in detail in A.4. For each electrode, layer, and lag relative to word onset, we used linear regression to estimate an encoding model that takes that layer's word embeddings as input and predicts the corresponding neural signals recorded by that electrode at that lag relative to word onset. To evaluate the linear model, we used the 50-dimensional weight vector estimated from the 9 training set folds to predict the neural signals in the held out test set fold. We repeated this for all folds to get predictions for all words at that lag. We evaluated the model's performance by computing the correlation between the predictions and true values for all words. (Fig. 1A). We repeated this process for all electrodes, for lags ranging from -2000 ms to +2000 ms (in 25ms increments) relative to word onset, and using the embeddings from each of the 48 hidden layers of GPT2-XL (Fig. 1B). The result for each electrode is a 48 x 161 matrix of correlations, where each row (corresponding to all lags for one layer) is what we call an encoding. We color-coded the encoding performance according to the index of the layer from which the embeddings were extracted, ranging from 1 (red) to 48 (blue; Fig. 1A). To evaluate our procedure on specific ROIs, we averaged the encodings over electrodes in the relevant ROIs. We then scaled the encoding model performance for each layer such that it peaks at 1; this allowed us to more easily visualize the temporal dynamics of encoding performance across layers.

## 4 GPT2-XL AS A MODEL FOR THE TEMPORAL HIERARCHY OF LANGUAGE COMPREHENSION IN THE INFERIOR FRONTAL GYRUS (IFG) - BROCA'S AREA

We started by focusing on neural responses for predictable words in electrodes at the inferior frontal gyrus (IFG), a central region for semantic and syntactic linguistic processing (Goldstein et al., 2022; Yang et al., 2019a; Hagoort & Indefrey, 2014; Hagoort, 2005; Ishkhanyan et al., 2020; LaPointe, 2013; Saur et al., 2008).

For each electrode in the IFG, we performed an encoding analysis for each GPT2-XL layer (1-48) at each lag (-2000 ms to 2000 ms in 25 ms increments). We then averaged encoding performance across all electrodes in the IFG to get a single mean encoding timecourse for each layer (all layers are plotted in Fig. 2C). We averaged over lags to get an average encoding performance per layer (Fig 2B). Significance was assessed using bootstrap resampling across electrodes (see A.7 in Supplementary materials and methods). The peak average correlation of the encoding models in the IFG was observed for the intermediate layer 22 (Fig. 2B; for other ROIs and predictability conditions, see Supp. Fig. 5). This corroborated recent findings from fMRI (Schrimpf et al., 2021; Toneva & Wehbe, 2019; Caucheteux et al., 2021) where encoding performance peaked in the intermediate

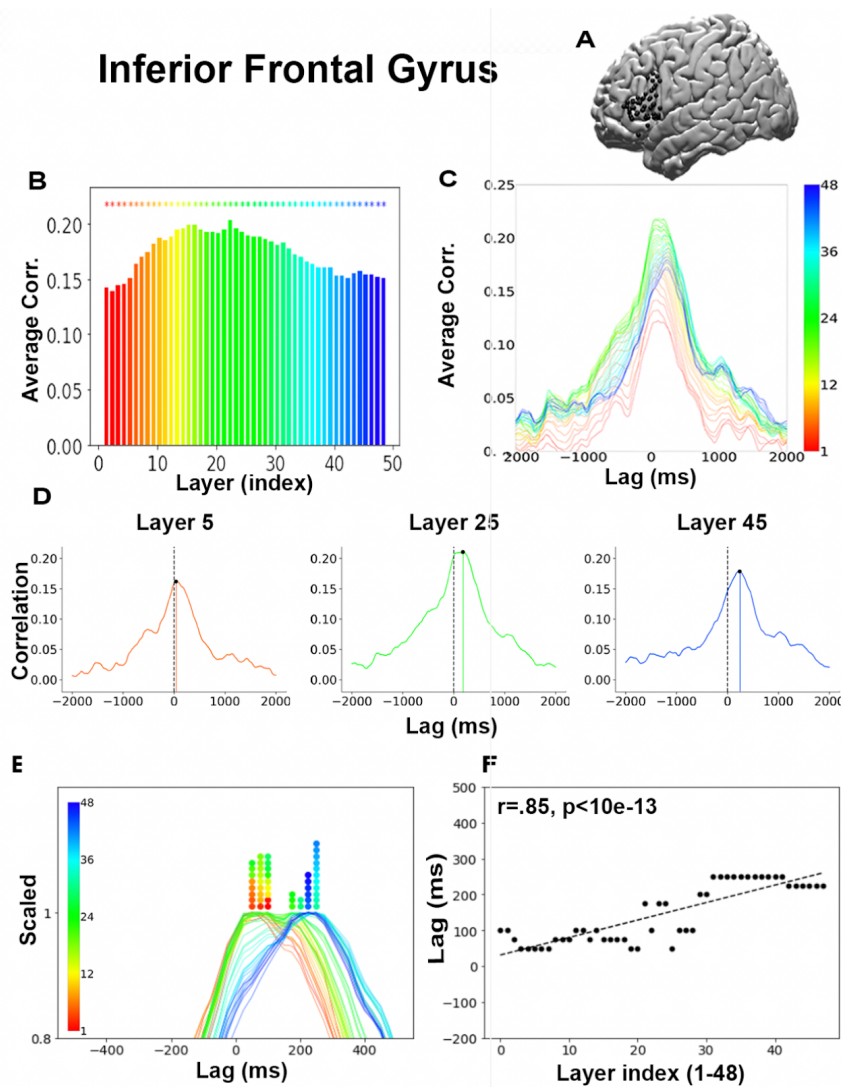

Figure 2: Temporal dynamics of layer-wise encoding for predictable words in IFG. (A) Location of IFG electrodes on the brain (black). (B) Average encoding performance across IFG electrodes and lags for each layer. (C ) Per-layer encoding plot in the IFG. (D) Encoding performance for layers 5, 25, and 45 showing the layer-wise shift of peak performance across lags. (E) Scaled encodings. (F) Scatter plot of the lag that yields peak encoding performance for each layer.

layers, yielding an inverted U-shaped curve across layers (Fig. 2B). This inverted U-shaped pattern held for all language areas (Supp. Fig. 5), suggesting that the layers of the model do not naively correspond to different cortical areas in the brain.

The fine-grained temporal resolution of ECoG recordings suggested a more subtle dynamic pattern. All 48 layers yielded robust encoding in the IFG, with encoding performance near zero at the edges of the 4000 ms window and increased performance around word onset. This can be seen in the combined plot of all 48 layers (Fig. 2C; for other ROIs and predictability conditions, see Supp. Fig. 6) and when we plotted individually selected layers (Fig. 2D, layers 5, 25, 45). A closer look at the encoding results over lags for each layer revealed an orderly dynamic in which the peak encoding performance for the early layers (e.g., layer 5, red, in Fig. 2D) tended to precede the peak encoding performance for intermediate layers (e.g., layer 25, green), which were followed by the later layers (e.g., layer 45, blue). To visualize the temporal sequence across lags, we normalized the encoding performance for each layer by scaling its peak performance to 1 (Fig. 2E; for other

ROIs and predictability conditions, see Supp. Fig. 7). From 2E, we observed that the layer-wise encoding models in the IFG tended to peak in an orderly sequence over time. To quantitatively test this claim, we correlated the layer index (1–48) with the lag that yielded the peak correlation for that layer (Fig. 2F). The analysis yielded a strong significant positive Pearson correlation of 0.85 (with p-value, p<10e-13). We decided to call the result of this procedure, when using Pearson correlation, the lag-layer correlation. Similar results were obtained with Spearman correlation; r = .80. We also conducted a non-parametric analysis where we permuted the layer index 100,000 times (keeping the lags that yielded the peak correlations fixed) while correlating the lags with these shuffled layer indices. Using the null distribution, we computed the percentile of the actual correlation (r=0.85) and got a significance of p<10e-5. To test the generalization of the effect across electrodes in the IFG, we ran the analysis on single electrodes (rather than on the average signal across electrodes) and then tested the robustness of the results across electrodes. For each electrode in the IFG we extracted the lags that yielded the maximal encoding performance for each layer in GPT2-XL. Next, we fit a linear mixed-effects model, using electrode as a random effect (model: max_lag $\sim$ 1 + layer + (1 + layer | electrode). The model converged, and we found a significant fixed effect of layer index (p < 10e-15). This suggested that the effect of layer generalizes across electrodes.

While we observed a robust sequence of lag-layer transitions across time, some groups of layers reached maximum correlations at the same temporal lag (ex. layers 1 and 2 in Fig. 2F). These non-linearities could be due to discontinuity in the match between GPT2-XL's 48 layers and transitions within the individual language areas. Alternatively, this may be due to the temporal resolution of our ECoG measurements, which, although high, were binned at 50 ms resolution. In other words, it is possible that higher-resolution ECoG data would disambiguate these layers.

We and others observed that the middle layers in DLMs better fit the neural signals than the early or late layers. In order to emphasize the relationship between the unique contributions of the intermediate representations induced by the layers and the dynamic of language comprehension in the brain, we projected out the embedding induced by the layer with the highest encoding performance (layer 22 in the IFG) from all other embeddings (induced by the other layers). We did this by subtracting from these embeddings their projection onto the embedding from layer 22. We then reran our encoding analysis. The results held even after controlling for the best-performing embedding (Supp. Fig. 8; for a full description of the procedure, see A.9.

Together, these results suggested that, for predictable words, the sequence of internal transformations across the layers in GPT2-XL matches the sequence of neural transformations across time within the IFG.

## 5 USING GPT2-XL TO RECOVER THE INCREASING TEMPORAL RECEPTIVE WINDOW ALONG THE LINGUISTIC PROCESSING HIERARCHY

We compared the temporal encoding sequence across three additional language ROIs (Fig. 3), starting with mSTG (near the early auditory cortex) and moving up along the ventral linguistic stream to the aSTG and TP. We did not observe obvious evidence for a temporal structure in the mSTG (Fig. 3, bottom right). This suggests that the temporal dynamic observed in IFG is regionally specific and does not occur in the early stages of the language processing hierarchy. We found evidence for the orderly temporal dynamic seen in the IFG in the aSTG (Pearson r = .92, p-value, p<10e-20) and TP (r = .93, p<10e-22). Similar results were obtained with Spearman correlation (mSTG r = -.24, p=.09; aSTG r=.94, p<10e-21; IFG r=.80, p<10e-11; TP r=.96, p<10e-27), demonstrating that the effect is robust to outliers. We followed our procedure for the IFG and conducted permutation tests by correlating 100,000 sets of layer indices with the true lags of peak encoding correlation for each layer. The resulting p-values were p<.02 for the mSTG, and p<10e-5 for the aSTG and IFG. To establish the relationship between layer order and latency across ROIs and electrodes, we ran a linear mixed model that, in addition to the fixed effects of layer and ROI, included electrode as a random effect (model: lag $\sim$ 1 + layer + ROI + (1 + layer | electrode)). All fixed effects were significant (p<.001), suggesting that the effect of layer generalizes across electrodes.

Our results suggested that neural activity in language areas proceeds through a nonlinear set of transformations that match the nonlinear transformations in deep language models. An alternative hypothesis is that the lag-layer correlation is due to a more rudimentary property of the network, in which early layers represent the previous word, late layers represent the current word, and intermedi-

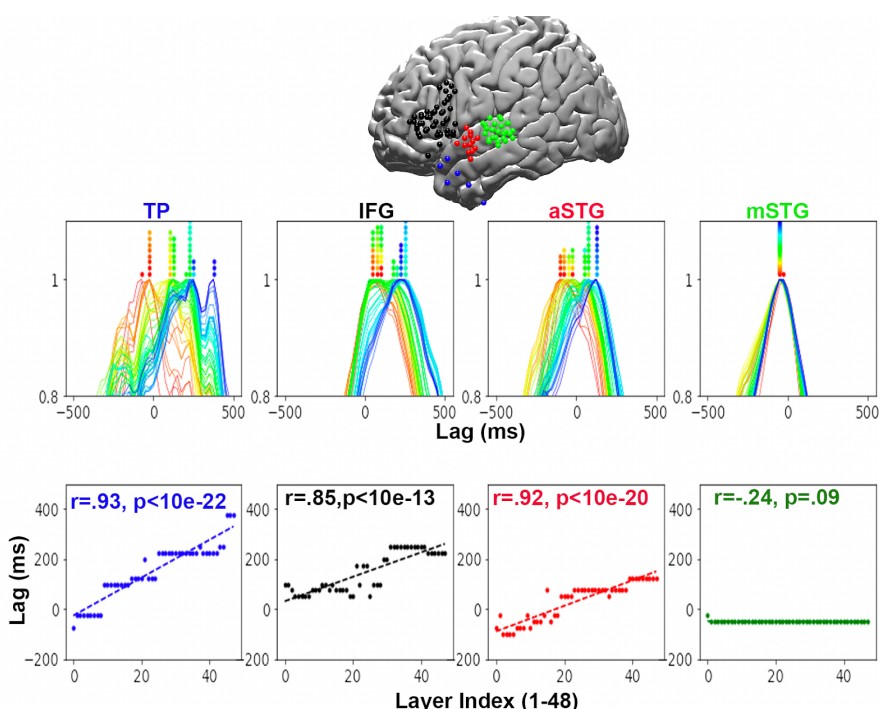

Figure 3: Temporal hierarchy along the ventral language stream for predictable words. (Top) Location of electrodes on the brain, color coded by ROI with blue, black, red, and green corresponding to TP, IFG, aSTG, and mSTG respectively. (Middle) Scaled encoding performance for these ROIs. (Bottom) Scatter plot of the lag that yields peak encoding performance for each layer.

ate layers carry a linear mix of both words. To test this alternative explanation, we designed a control analysis where we sampled 46 intermediate "pseudo-layers" from a set of $\sim 10^3$ embeddings that we generated by linearly interpolating between the first and last layers. We repeated this process $10^4$ times (see A.10), and for each set, we sorted the embeddings and computed the lag-layer correlation. Supplementary figure 9 plots the slopes obtained for the controlled linear transformations versus the actual nonlinear transformations. The results indicated that the actual lag-layered correlations were significantly higher than the ones achieved by the linearly-interpolated layers (p<.01). This indicated that GPT2-XL, with its non-linear transformations captured the brain dynamic better than a simpler model that performed a linear transformation between the embeddings of the previous and current words.

We then explored the timescales of the temporal progression in different brain areas. It seemed that it gradually increased along the ventral linguistic hierarchy (see the increase in steepness of the slopes across language areas in Fig. 3 as you move right to left). In order to evaluate this we computed standard deviations of the encoding maximizing lags within ROIs. We used Levene's test and found significant differences between these standard deviations for the mSTG and aSTG (F = 48.1, p<.01) and for the aSTG and TP (F = 5.8, p<.02). The largest within-ROI temporal separation across layer-based encoding models was seen in the TP, with more than a 500 ms difference between the peak for layer 1 (around -100 ms) and the peak for layer 48 (around 400 ms).

## 6 DISCUSSION

In this work we leverage the superior temporal resolution of ECoG, and all 48 layers of GPT2-XL to gain insights into the subtleties of language comprehension in the human brain. We found that the layer-wise transformations learned by GPT2-XL map onto the temporal sequence of transformations of natural language in high-level language areas. This finding reveals an important link between how DLMs and the brain process language: conversion of discrete input into multidimensional (vectorial)

embeddings, which are further transformed via a sequence of nonlinear transformations to match the context-based statistical properties of natural language (Manning et al., 2020). These results provide additional evidence for shared computational principles in how DLMs and the human brain process natural language.

Our study points to implementation differences between the internal sequence of computations in transformer-based DLMs and the human brain. GPT2-XL relies on a "transformer", a neural network architecture developed to process hundreds to thousands of words in parallel during its training. In other words, transformers are designed to parallelize a task largely computed serially, word by word, in the human brain. While transformer-based DLMs process words sequentially over layers, in the human brain, we found evidence for similar sequential processing but over time relative to word onset within a given cortical area. For example, we found that within high-order language areas (such as IFG and TP), a sequence of temporal processing corresponded to the sequence of layer-wise processing in DLMs. In addition, we demonstrated that this correspondence is a result of the non-linear transformations across layers in the language model and is not a result of straightforward linear interpolation between the previous and current words (Supp Fig. 9).

The implementation differences between the brain and language model may suggest that cortical computation within a given language area is better aligned with recurrent architectures, where the internal computational sequence is deployed over time rather than over layers. This sequence of temporal processing unfolds over longer timescales as we proceed up the processing hierarchy, from aSTG to IFG and TP. Second, it may be that the layered architecture of GPT2-XL is recapitulated within the local connectivity of a given language area like IFG (rather than across cortical areas). That is, local connectivity within a given cortical area may resemble the layered graph structure of GPT2-XL. Third, it is possible that long-range connectivity between cortical areas could yield the temporal sequence of processing observed within a single cortical area. Together, these results hint that a deep language model with stacked recurrent networks may better fit the human brain's neural architecture for processing natural language. Interestingly, several attempts have been made to develop new neural architectures for language learning and representation, such as universal transformers (Dehghani et al., 2018; Lan et al., 2020) and reservoir computing (Dominey, 2021). Future studies will have to compare how the internal processing of natural language compares between these models and the brain.

Another fundamental difference between deep language models and the human brain is the characteristics of the data used to train these models. Humans do not learn language by reading text. Rather they learn via multi-modal interaction with their social environment. Furthermore, the amount of text used to train these models is equivalent to hundreds (or thousands) of years of human listening. An open question is how DLMs will perform when trained on more human-like input: data that are not textual but spoken, multimodal, embodied and immersed in social actions. Interestingly, two recent papers suggest that language models trained on more realistic human-centered data can learn a language as children do (Huebner et al., 2021; Hosseini et al., 2022). However, additional research is needed to explore these questions.

Finally, this paper provides strong evidence that DLMs and the brain process language in a similar way. Given the clear architectural differences between DLMs and the human brain, the convergence of their internal computational sequences may be surprising. Classical psycholinguistic theories postulated an interpretable rule-based symbolic system for linguistic processing. In contrast, DLMs provide a radically different framework for learning language through its statistics by predicting speakers' language use in context. This kind of unexpected mapping (layer sequence to temporal sequence) can point us in novel directions for both understanding the brain and developing neural network architectures that better mimic human language processing. This study provides strong evidence for shared internal computations between DLMs and the human brain and calls for a paradigm shift from a symbolic representation of language to a new family of contextual embeddings and language statistics-based models.

## 7 REPRODUCIBILITY STATEMENT

We describe electrode preprocessing in detail in A.2. Embedding extraction and preprocessing is described in A.3 and A.4. We provide a detailed description of our encoding model in 3.2 and

A.5.We discuss how to generate our encoding and lag-layer plots in 4. The code used to produce our results will be made available upon publication.

## 8    ETHICS STATEMENT

All patients volunteered for this study, and according to the host institution's Institutional Review Board, all participants had elected to undergo intracranial monitoring for clinical purposes and provided oral and written informed consent before study participation.

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

## A  SUPPLEMENTARY MATERIALS AND METHODS

### A.1  DATA ACQUISITION

Ten patients (5 female; 20–48 years old) listened to the same story stimulus ("So a Monkey and a Horse Walk Into a Bar: Act One, Monkey in the Middle") from beginning to end. The audio narrative is 30 minutes long and consists of 5000 words. One patient was removed from further analyses due to excessive epileptic activity and low SNR across all experimental data collected during the day. All patients volunteered for this study and according to the host institution's Institutional Review Board, all participants had elected to undergo intracranial monitoring for clinical purposes and provided oral and written informed consent before study participation. Language areas were localized to the left hemisphere in all epileptic participants using the WADA test. All epileptic participants were tested for verbal comprehension index (VCI), perceptual organization index (POI), processing speed index (PSI), and Working Memory Index (VMI). See the supplementary table 1, which summarizes each patient's pathology and neuropsychological scores. In addition, all patients passed the Boston

Picture Naming Task and auditory naming task (Tombaugh & Hubiey, 1997; Hamberger & Tamny, 1999). Due to the lack of significant language deficits in the participants, our results will generalize outside of our cohort.

Patients were informed that participation in the study was unrelated to their clinical care and that they could withdraw from the study at any point without affecting their medical treatment. After consenting to participate in the experiment, they were told they would hear a 30-minute podcast and were asked to listen to it.

For each patient, electrode placement was determined by clinicians based on clinical criteria. One patient consented to have an FDA-approved hybrid clinical-research grid implanted, which includes standard clinical electrodes and additional electrodes between clinical contacts. The hybrid grid provides a higher spatial coverage without changing clinical acquisition or grid placement. Across all patients, 1106 electrodes were placed on the left hemisphere and 233 on the right hemisphere. Brain activity was recorded from a total of 1339 intracranially implanted subdural platinum-iridium electrodes embedded in silastic sheets (2.3 mm diameter contacts, Ad-Tech Medical Instrument; for the hybrid grids 64 standard contacts had a diameter of 2 mm and additional 64 contacts were 1 mm diameter, PMT corporation, Chanassen, MN). Decisions related to electrode placement and invasive monitoring duration were determined solely on clinical grounds without reference to this or any other research study. Electrodes were arranged as grid arrays (8 × 8 contacts, 10 or 5 mm center-to-center spacing), or linear strips.

Pre-surgical and post-surgical T1-weighted MRIs were acquired for each patient, and the location of the electrodes relative to the cortical surface was determined from co-registered MRIs or CTs following the procedure described in Yang et al. (2012). Co-registered, skull-stripped T1 images were nonlinearly registered to an MNI152 template and electrode locations were then extracted in Montreal Neurological Institute (MNI) space (projected to the surface) using the co-registered image. All electrode maps are displayed on a surface plot of the template, using an Electrode Localization Toolbox for MATLAB.

## A.2 PREPROCESSING

66 electrodes from all patients were removed due to faulty recordings. Large spikes in the electrode signals, exceeding four quartiles above and below the median, were removed, and replacement samples were imputed using cubic interpolation. We then re-referenced the data to account for shared signals across all electrodes using the Common Average Referencing (CAR) method or an ICA-based method (based on the participant's noise profile). High-frequency broadband (HFBB) power provided evidence for a high positive correlation between local neural firing rates and high gamma activity. Broadband power was estimated using 6-cycle wavelets to compute the power of the 70-200 Hz band (high-gamma band), excluding 60, 120, and 180 Hz line noise. Power was further smoothed with a Hamming window with a kernel size of 50 ms.

## A.3 LINGUISTIC EMBEDDINGS

To extract contextual embeddings for the stimulus text, we first tokenized the words for compatibility with GPT2-XL. We then ran the GPT2-XL model implemented in HuggingFace (4.3.3) (Wolf et al., 2020) on this tokenized input. To construct the embeddings for a given word, we passed the set of up to 1023 preceding words (the context) along with the current word as input to the model. The embedding we extract is the output generated for the previous word. This means that the current word is not used to generate its own embedding and its context only includes previous words. We constrain the model in this way because our human participants do not have access to the words in the podcast before they are said during natural language comprehension.

GPT2-XL is structured as a set of blocks that each contain a self-attention sub-block and a subsequent feedforward sub-block. The output of a given block is the summation of the feedforward output and the self-attention output through a residual connection. This output is also known as a "hidden state" of GPT2-XL. We consider this hidden state to be the contextual embedding for the block that precedes it. For convenience, we refer to the blocks as "layers"; that is, the hidden state at the output of block 3 is referred to as the contextual embedding for layer 3. To generate the contextual embeddings for each layer, we store each layer's hidden state for each word in the input text.

Fortunately, the HuggingFace implementation of GPT2-XL automatically stores these hidden states when a forward pass of the model is conducted. Different models have different numbers of layers and embeddings of different dimensionality. The model used herein, GPT2-XL, has 48 layers, and the embeddings at each layer are 1600-dimensional vectors. For a sample of text containing 101 words, we would generate an embedding for each word at every layer, excluding the first word as it has no prior context. This results in 48 1600-dimensional embeddings per word and with 100 words; 48 * 100 = 4800 total 1600-long embedding vectors. Note that in this example, the context length would increase from 1 to 100 as we proceed through the text.

## A.4 DIMENSIONALITY REDUCTION

Before fitting the encoding models, we first reduce the dimensionality of the embeddings from each layer separately by applying principal component analysis (PCA) and retaining the first 50 components. This procedure effectively focuses our subsequent analysis on the 50 orthogonal dimensions in the embedding space that account for the most variance in the stimulus. We do not compute PCA on the entire set of embeddings (layers x words) as that would result in mixing information between layers. However, in our main results, we computed PCA on the concatenation of train and test folds for a given layer and predictability condition. In order to verify our results are not impacted by leakage between train and test data, we fit PCA only on the training folds, and used this projection matrix on the test fold (for each test fold we learned a separate PCA projection matrix). The results were reproduced and are reported in Supp Fig. 10.

## A.5 ENCODING MODELS

Linear encoding models were estimated at each lag (-2000 ms to 2000 ms in 25-ms increments) relative to word onset (0 ms) to predict the brain activity for each word from the corresponding contextual embedding. Before fitting the encoding model, we smoothed the signal using a rolling 200-ms window (i.e., for each lag the model learns to predict the average signal +-100 ms around the lag). For the effect of the window size on the results see Supp. Fig. 11. We used a 10-fold cross-validation procedure ensuring that for each cross-validation fold, the model was estimated from a subset of training words and evaluated on a non-overlapping subset of held-out test words: the words and the corresponding brain activity were split into a training set (90% of the words) for model estimation and a test set (10% of the words) for model evaluation. Encoding models were estimated separately for each electrode (and each lag relative to word onset). For each cross-validation fold, we used ordinary least squares (OLS) multiple linear regression to estimate a weight vector (50 coefficients for the 50 principal components) based on the training words. We then used those weights to predict the neural responses at each electrode for the test words. We repeated this for the remaining test folds to obtain results for all words. We evaluated model performance by computing the correlation between the predicted brain activity at test time and the actual brain activity, across words (for a given lag); we then averaged these correlations across electrodes within specific ROIs. This procedure was performed for all the layers in GPT2-XL to generate an "encoding" for each layer.

## A.6 PREDICTABLE AND UNPREDICTABLE WORDS

After generating encodings for all words in the podcast transcript, we split the embeddings into two subsets: words that the model was able to predict and words that the model was unable to predict. A word was considered to be predictable if the model assigned that word the highest probability of occurring next among all possible words. We refer to this subset of embeddings as "top 1 predictable" or just "predictable" (1709 words out of 4744 = 36%). To reduce the stringency of top 1 prediction, we also created subsets of "top 5 predictable" (2936 words out of 4744 = 62%) and "top 5 unpredictable" words where the criterion for predictability was that the probability assigned by the model to predictable words must be among the highest five probabilities assigned by the model. Top 5 unpredictable words, which we also refer to as simply "unpredictable" words were not in that subset. We then trained linear encoding models as outlined above on the sets of top 1 predictable and top 5 unpredictable embeddings.

## A.7 STATISTICAL SIGNIFICANCE

To establish the significance of the bars in Fig. 2B we conducted a bootstrapping analysis for each lag. Given the values of the electrodes in a specific layer and a specific ROI, we sampled the values of the max correlations with replacement ($10^4$ samples including values for all electrodes). For each sample, we computed the mean and generated a distribution (consisting of $10^4$ points). We then compared the actual mean for the lag-ROI pair to estimate how significant it is given the generated distributions. The results can be seen in 5. The '*' indicates a two-tailed significance of p<0.01.

## A.8 ELECTRODE SELECTION

To identify significant electrodes, we used a nonparametric statistical procedure with correction for multiple comparisons (Nichols & Holmes, 2001). At each iteration, we randomized each electrode's signal phase by sampling from a uniform distribution. This disconnected the relationship between the words and the brain signal while preserving the autocorrelation in the signal. We then performed the encoding procedure for each electrode (for all lags). We repeated this process 5000 times. After each iteration, the encoding model's maximal value across all lags was retained for each electrode. We then took the maximum value for each permutation across electrodes. This resulted in a distribution of 5000 values, which was used to determine the significance for all electrodes. For each electrode, a p-value was computed as the percentile of the non-permuted encoding model's maximum value across all lags from the null distribution of 5000 maximum values. Performing a significance test using this randomization procedure evaluates the null hypothesis that there is no systematic relationship between the brain signal and the corresponding word embedding. This procedure yielded a p-value per electrode, corrected for the number of models tested across all lags within an electrode. To further correct for multiple comparisons across all electrodes, we used a false-discovery rate (FDR). Electrodes with q-values less than .01 are considered significant. This procedure identified 160 electrodes in early auditory areas, motor cortex, and language areas in the left hemisphere. We used subsets of this list corresponding to the IFG (n=46), TP (n=6), aSTG (n=13) and mSTG (n=28).

## A.9 CONTROLLING FOR THE BEST-PERFORMING LAYER

We divided our results into all combinations of predictable, unpredictable, all words x mSTG, TP, aSTG, IFG and found the layer with maximum encoding correlation (highest peak in the unnormalized encoding plot) for each combination (max-layer) (Supp. Table 2). For each layer, we projected its embeddings onto the embeddings for the max-layer using the dot product (separate projection per word). We then subtracted these projections from the layer's embeddings to get a set of embeddings for that layer that are orthogonal to their counterparts in the max-layer. We also project the max-layer from itself to ensure that the information was removed properly (seen in black in Supp. Fig. 8). We then ran encoding on these sets of embeddings as normal (Supp. Fig. 8 for the IFG result). Our finding of a temporal ordering of layer-peak correlations in the IFG is preserved.

## A.10 INTERPOLATION SIGNIFICANCE TEST

To show that the contextual embeddings generated through the layered computations in GPT2 are significantly different from those generated through a simple linear interpolation between the input layer (previous word) and output layer (current word), we linearly interpolated $\sim 10^3$ embeddings between the first and last contextual embeddings of GPT2-XL. We then re-ran our lag layer analysis for 10,000 iterations (for each ROI x predictability condition), except instead of using the 48 layers of GPT2-XL, we used the first and last layers, and 46 intermediate layers, sampled without replacement from the set of linear interpolations and then sorted. We constructed a distribution of correlations between layer index (the sampled layers were sorted and assigned indices 2-47) and the corresponding lags that maximize the encodings for each layer. We then computed the p-value of our true correlation for that ROI x word classification condition concerning this distribution. The results can be seen in Supp. Fig. 9.

### A.11 UNPREDICTABLE WORDS

The temporal correspondence described in the main text was observed for words the model accurately predicted; does the same pattern hold for words that were not accurately predicted? We conducted the same layer-wise encoding analyses in the same ROIs for unpredictable words—i.e., words for which the probability assigned to the word was not among the top-5 highest probabilities assigned by the model (N = 1808). For these results, see the unpredictable column in Supp. Figs 5, 6, 7. We still see evidence, albeit slightly weaker, for layer-based encoding sequences in the IFG (r = .81, p<10e-11) and TP (r = .57, p<10e-4), but not aSTG (r = .09, p>.55) or mSTG (r = -.10,p>.48). Similar results were obtained with Spearman correlation (mSTG r = -.10, p>.48; aSTG r=.02, p>.9; IFG r=.8, p<10e-11; TP r=.72, p<10e-8), demonstrating that the effect is robust to outliers. We conducted permutation tests that yielded the following p-values: p>.24 (mSTG), p>.27 (aSTG), and p<10e-5 (TP, IFG). While we observed a sequence of temporal transitions across layers in language areas, we did not observe such transformations in mSTG. The lack of temporal sequence in mSTG may be due to the fact that it is sensitive to speech-related phonemic information rather than word-level linguistic analysis (Liebenthal & Bernstein, 2017; Oganian & Chang, 2019; Chechik & Nelken, 2012; Farbood et al., 2015).

We noticed a crucial difference between the encoding of the predictable and unpredictable words in the IFG. In the IFG, the encoding for early layers (red) shifted from around word onset (lag 0) for predictable words to later lags (around 300ms) for unpredictable words. We ran a paired t-test to compare the average of lags (over the electrodes in an ROI) that yield the maximal correlations (i.e., peak encoding performance) across predicted and unpredicted words for each layer. The paired t-test indicated that the shift of the lag of peak encoding (at the ROI level) was significant for 9 out of the 12 first layers (corrected for multiple comparisons, see Supp. Table 3, q<0.01).

### A.12 COMPUTE

This work employed two computationally intensive processes. These were the generation of GPT2-XL embeddings, and the training of our encoding models. For the former process we allocated 1 GPU and 4 CPUs, 192GB of memory and 6 hours of time. To train encoding models for all electrodes and lags we allocated 4 CPUs, and 30GB of memory. This process took only 3 minutes, but had to be repeated for all layers and predictability conditions. To reproduce the core analyses in the main body of our paper, with the 48 layers and predictable words, would therefore have taken around 2.5 hours. The computational resources used were on a shared cluster at the institution where the computational aspects of this work were conducted.

## B FIGURES AND TABLES

Table 1: Additional information about patient pathology and neuropsychological scores.

| NO. | Neuropsych Score | Pathology/Epilepsy type/Seizure Focus | Implant |
|---|---|---|---|
| 1 | VCI: 145 POI: 96 PSI: 86 WMI: 95 | Focal epilepsy arising from the left hemisphere with a broad focus involving the left temporal neocortex (superior, middle, inferior temporal gyri) left frontal operculum, inferior postcentral gyrus, insula | Left grid and strips |
| 2 | VCI: 87 POI: 123 PSI: 97 WMI: 89 | Left temporal lobe epilepsy. Ictal onsets localized to the left temporal lobe (perilesionally) and left posterior mesial temporal lobe. | Left grid, strips, and depth electrodes |
| 3 | VCI: 100 POI: 109 PSI: 92 WMI: 86 | Focal epilepsy localized to the left posterior insula and periopercular region at the frontoparietal junction. | Left grid, strips, and depth electrodes |
| 4 | not found | Probable focal epilepsy, not clearly lateralized or localized. ICEEG showed definitively that disabling clinical events were psychogenic non-epileptic attacks. | Left grid, strips, and depth electrodes |
| 5 | VCI: 96 POI: 79 PSI: 81 WMI: 86 | Left hemispheric multilobar epilepsy | Left grid, strips, and depth electrodes |
| 6 | not found | Bilateral mesial temporal lobe epilepsy | Bilateral strips and depth electrodes |
| 7 | VCI: 107 POI: 104 PSI: 111 WMI: 114 | Right anteromesial temporal lobe epeilpsy. ICEEG localized ictal onsets to the right temporal pole and right hippocampus | Bilateral strips and depths, and a left grid |
| 8 | VCI: 72 POI: 88 WMI: 71 PSI: 89 | Probable left frontal lobe epilepsy. ICEEG did not definitively localize ictal onset within the left hemisphere; the first electrographic changes were most consistent with spread pattern. The working diagnosis is a left frontal focus. | Left depths and strips |

Table 2: Layers that maximize encoding performance for different combinations of ROI and word classification

|  | PREDICTABLE | UNPREDICTABLE | ALL |
|---|---|---|---|
| mSTG | 22 | 26 | 22 |
| TP | 22 | 26 | 22 |
| aSTG | 17 | 22 | 17 |
| ifg | 24 | 20 | 16 |

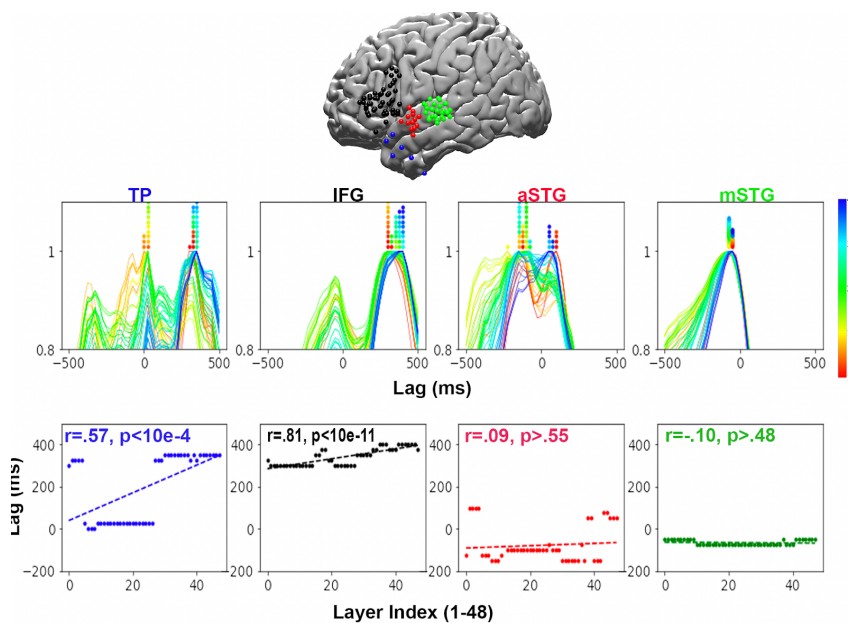

Figure 4: Temporal hierarchy along the ventral language stream for unpredictable words. (Top) location of electrodes on the brain, color coded by roi with blue, black, red, green corresponding to TP, IFG, aSTG and mSTG respectively. (Middle) Scaled encoding performance for these ROIs. (Bottom) Scatter plot of the lag that yields peak encoding performance for each layer.

Table 3: The p-value and FDR-corrected q-value of the paired sampled t-test comparing the lags that achieve maximal correlation in the encoding across the different layers (n=48) of GPT2-XL.

| LAYER INDEX | P-VALUE | Q-VALUE | LAYER INDEX | P-VALUE | Q-VALUE |
|---|---|---|---|---|---|
| 1 | 0.784826 | 0.459996 | 25 | 0.731631 | 0.3963 |
| 2 | 0.061834 | 0.015458 | 26 | 0.719935 | 0.360514 |
| 3 | 0.016337 | 0.000953 | 27 | 0.608419 | 0.278859 |
| 4 | 0.016337 | 0.00111 | 28 | 0.569881 | 0.249323 |
| 5 | 0.409457 | 0.153546 | 29 | 0.38613 | 0.136755 |
| 6 | 0.016337 | 0.001688 | 30 | 1 | 0.949051 |
| 7 | 0.016337 | 0.001847 | 31 | 0.990003 | 0.696491 |
| 8 | 0.035199 | 0.0066 | 32 | 1 | 0.906098 |
| 9 | 0.016522 | 0.002409 | 33 | 1 | 0.94135 |
| 10 | 0.023182 | 0.003864 | 34 | 1 | 0.922248 |
| 11 | 0.016337 | 0.002042 | 35 | 1 | 0.9526 |
| 12 | 0.051168 | 0.01066 | 36 | 0.719935 | 0.374966 |
| 13 | 0.283744 | 0.094581 | 37 | 0.927577 | 0.618385 |
| 14 | 0.276009 | 0.086253 | 38 | 1 | 0.844096 |
| 15 | 0.21497 | 0.0627 | 39 | 0.784826 | 0.474165 |
| 16 | 0.059726 | 0.013687 | 40 | 0.927577 | 0.61807 |
| 17 | 0.016337 | 0.000372 | 41 | 0.820915 | 0.513072 |
| 18 | 0.191238 | 0.051794 | 42 | 1 | 0.961332 |
| 19 | 0.425111 | 0.168273 | 43 | 1 | 0.736217 |
| 20 | 0.990003 | 0.701252 | 44 | 1 | 0.942203 |
| 21 | 1 | 0.993479 | 45 | 0.548249 | 0.228437 |
| 22 | 0.749748 | 0.421733 | 46 | 1 | 0.968246 |
| 23 | 0.703456 | 0.337073 | 47 | 1 | 1 |
| 24 | 1 | 0.825196 | 48 | 1 | 0.907848 |

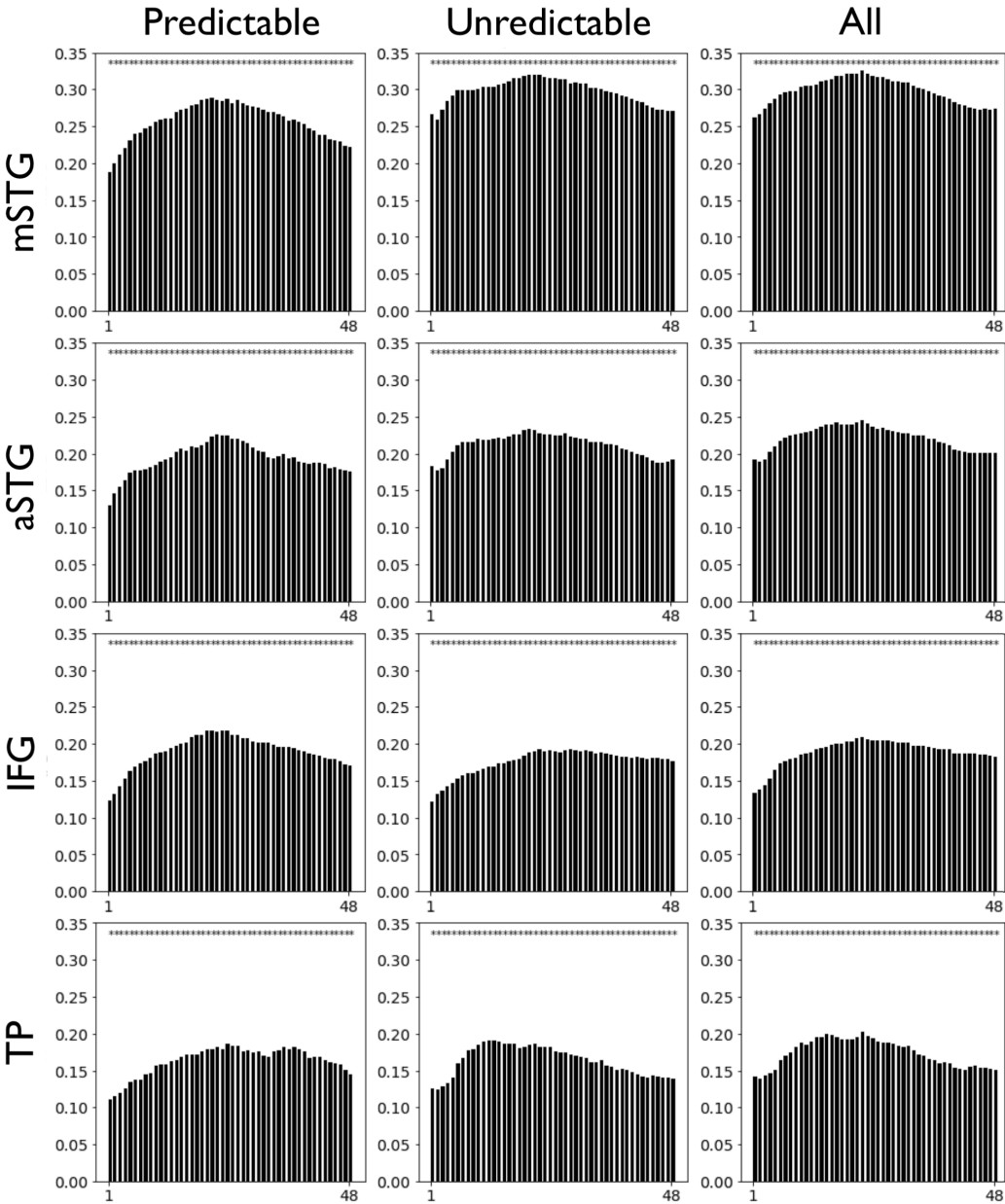

Figure 5: Peak correlations of electrode-averaged encodings for each combination of layer (1-48), brain area (mSTG, aSTG, IFG and TP) and word classification (predictable, unpredictable, all words). The significance test is done using a bootstrap analysis across the electrodes.

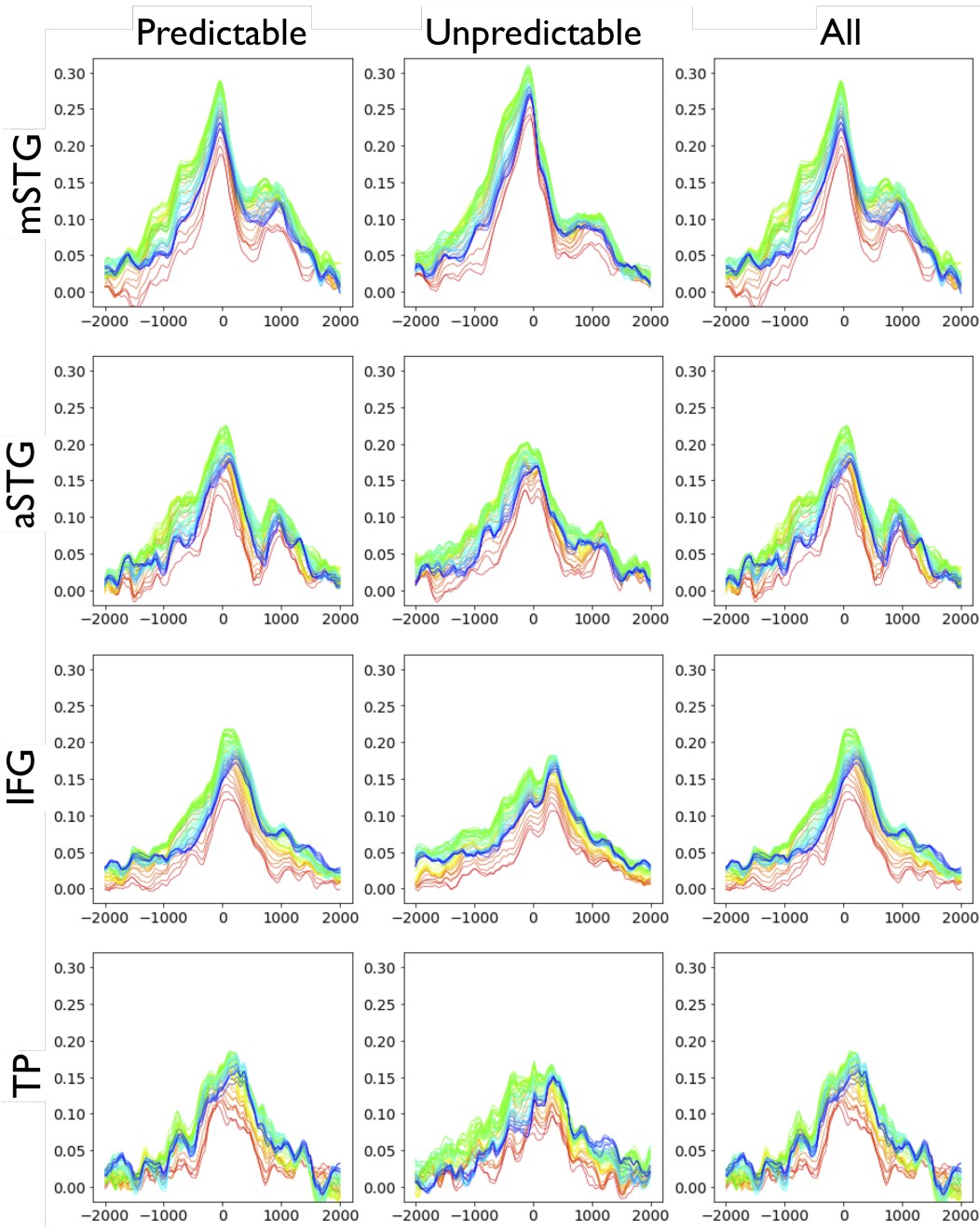

Figure 6: Encoding averaged over electrodes for each combination of layer (1-48), brain area (mSTG, aSTG, IFG and TP) and word classification (predictable, unpredictable, all words).

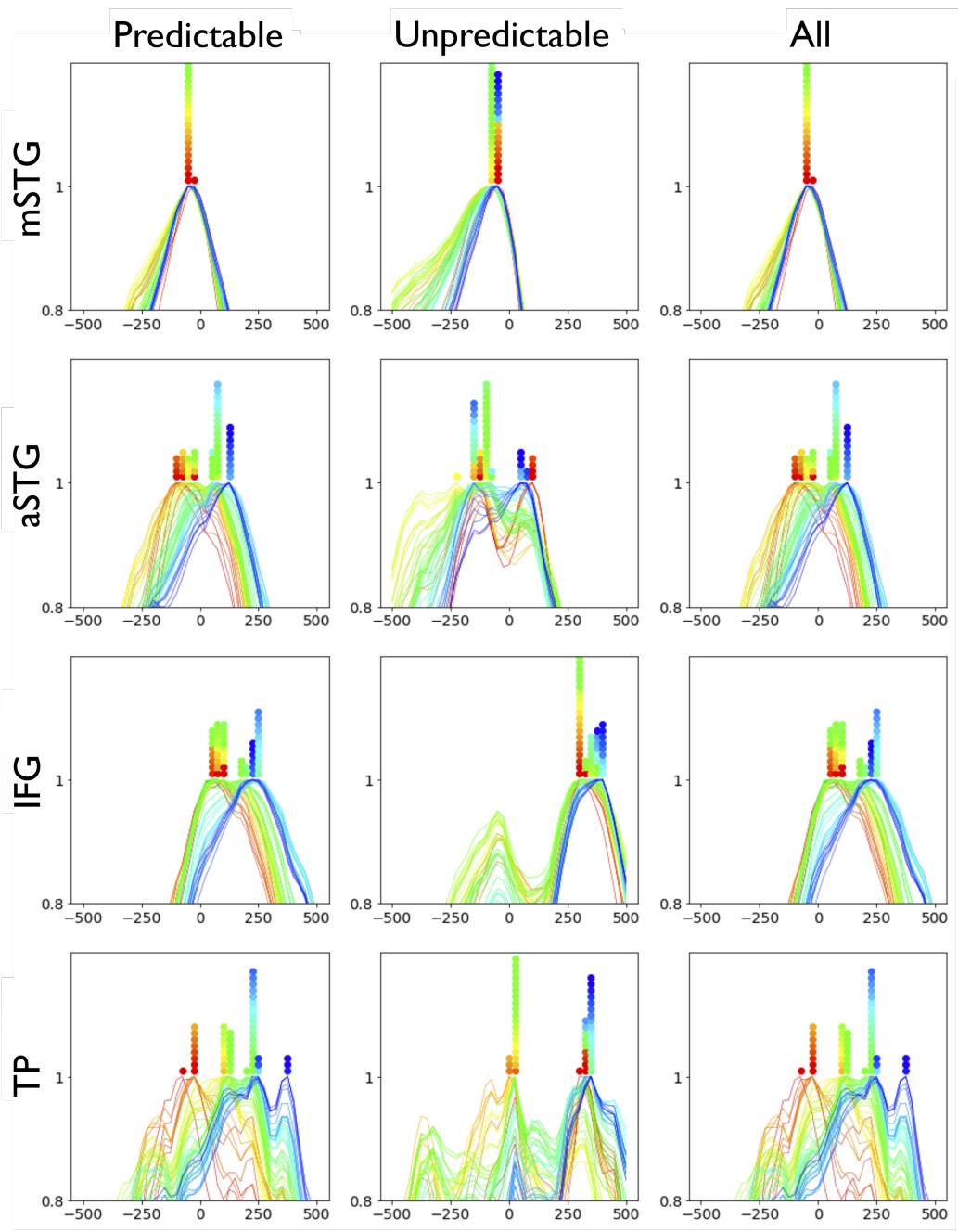

Figure 7: Scaled encoding for each combination of layer (1-48), brain area (mSTG, aSTG, IFG and TP) and word classification (predictable, unpredictable, all words). For completion the correlation between the layer index and max-lag for condition 'All': mSTG (r=.56, p < 10e-4), aSTG ( r=.81, p<10e-11), IFG ( r=.89 ,p<10e-16), TP ( r=.75 ,p<10e-9)

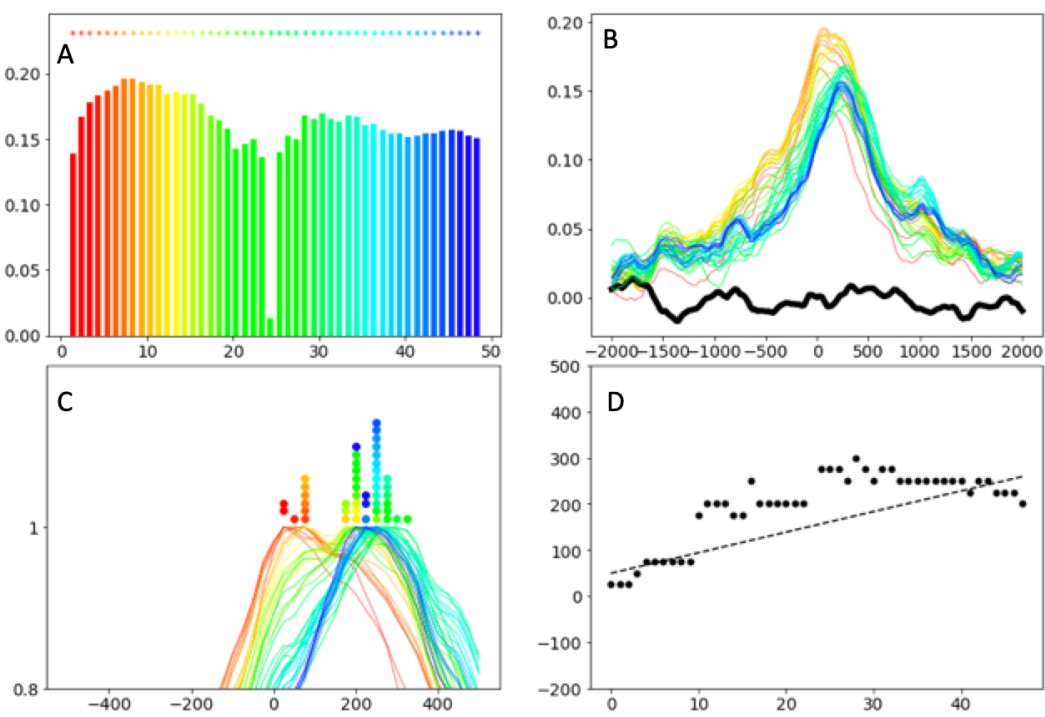

Figure 8: Replicating the findings regarding the IFG controlling for the correlation between the optimal layer (layer 22) and the other layers. We projected the best-performing embedding out of the embeddings at all other layers, then repeated our analyses. (A) Correlation between the brain and different embeddings is preserved after controlling for the variance explained in other layers by the optimal layer. (B–D). The temporal relation between the optimal encoding performance for lag and layer index is also preserved after controlling for the correlation between the optimal layer and other layers.

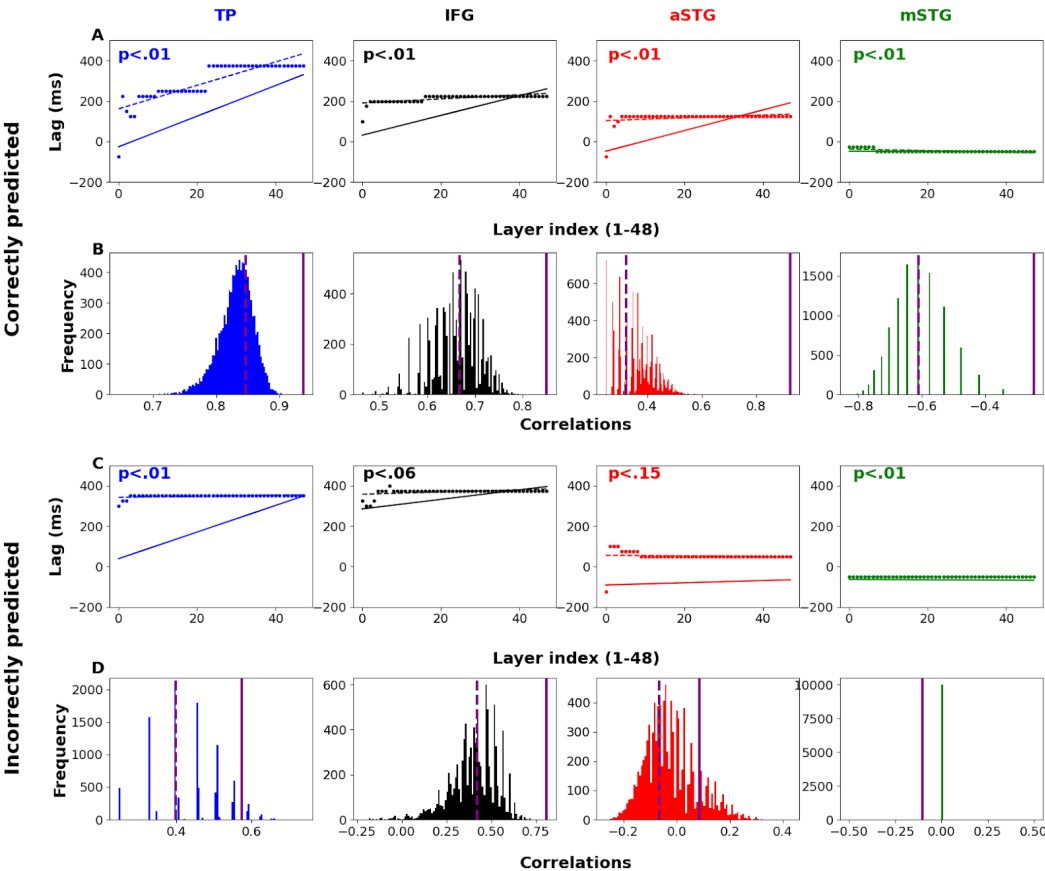

Figure 9: Control analysis for the alternative hypothesis that the lag-layer correlation we observe is due to a more rudimentary network property in which early layers represent the previous word, late layers represent the current word, and intermediate layers carry a linear interpolation between these words. (A) The lag-layer slope of the even-spaced pseudo layers (dashed line) versus the actual lag-layer analysis (solid line) for predictable words. The p-value is calculated from B. (B) Distribution of correlations induced by 10,000 iterations of sampling pseudo interpolated layers. The horizontal dashed purple line is the correlation achieved in the even-space case, and the continuous purple line represents the values achieved by the actual correlation. Importantly, the p-values are smaller than .01. For completion, we present the same analyses and plots for unpredictable words in C & D.

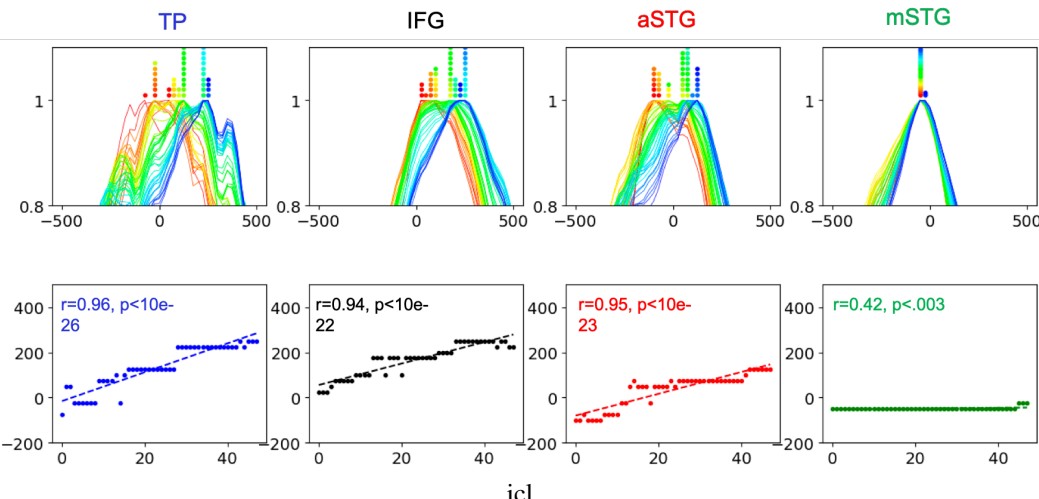

Figure 10: In our main results we trained PCA on the entire set of embeddings for a given layer and predictability condition (training and test folds) and used the resulting projection matrix to reduce the embedding size to 50. To verify our results still hold without leakage, we trained PCA on only the training folds and used the resulting projection matrix to reduce the embedding size to 50 (we repeated this process for each of the 10 training folds). On top we show the scaled encodings for each ROI and on the bottom we show the relationship between layer and lag of peak encoding correlation.

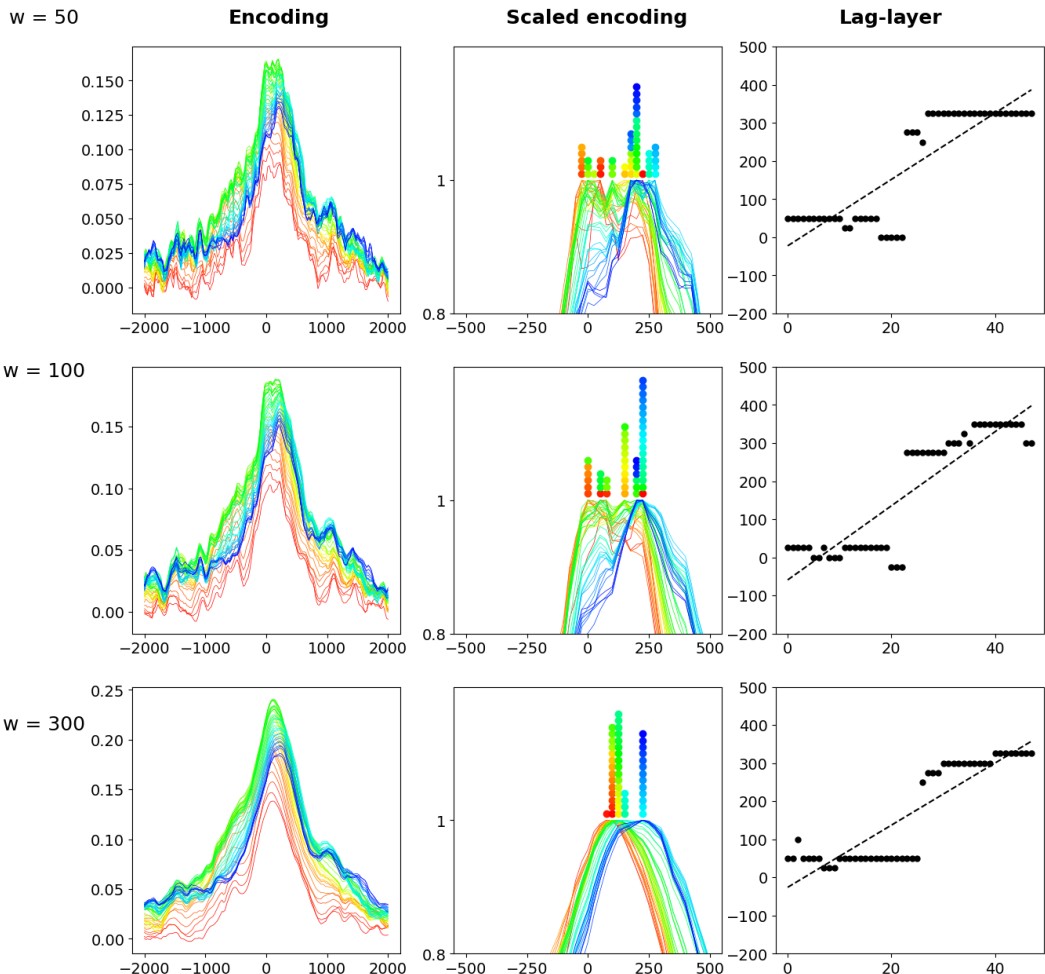

Figure 11: We re-ran the encoding analysis, scaled encoding analysis and lag-layer analysis for different smoothing window sizes (windows of 50, 100, 300). The correlations are all positive ( r>0.75) and significant (p<1e-10)

