# OpenReview forum: "The Temporal Structure of Language Processing in the Human Brain Corresponds to The Layered Hierarchy of Deep Language Models"
_ICLR.cc/2024/Conference — Submitted to ICLR 2024_

### Official Review · Reviewer_4Wkk · 2023-10-28

**Soundness:** 3 good
**Presentation:** 3 good
**Contribution:** 3 good
**Rating:** 6
**Confidence:** 3

**Summary:**

- The paper investigates how deep language models (DLMs) like GPT-2 map onto the spatiotemporal dynamics of language processing in the human brain.
- The authors use ECoG to record neural activity during narrative listening and compare embeddings from GPT-2 layers to predict brain activity over time.
- The main finding is that early layers predict early brain activity while later layers predict later brain activity, suggesting DLM computations mirror the temporal accumulation of linguistic information in the brain.

**Strengths:**

1. The use of ECoG provides higher spatiotemporal resolution compared to fMRI, allowing the authors to study language processing dynamics at a more fine-grained level.

2. Analyzing all 48 layers of GPT-2 is more comprehensive than prior work that looked at just the final layer. This enables new findings about how early vs late layers map to brain activity over time.

3. The authors employ rigorous statistical analyses, including permutation testing and linear mixed effects modeling, to validate the layer-timing relationships across electrodes and ROIs.

**Weaknesses:**

1. While the authors state they used 10 patients, details are lacking on the specific demographic and clinical characteristics of the patient sample. This could impact the generalizability of the findings.

2. Preprocessing steps for the ECoG data should be described in more detail (e.g. filtering, re-referencing, artifact rejection).

3. The encoding model parameters, such as context length, smoothing windows, regularization, could be optimized more thoroughly. Ablations could be performed to test the effect of these modeling choices.

4. The linear interpolation analysis addresses one type of baseline model, but comparisons to other neural language models (BERT, ELMo etc) would be informative.

5. The theoretical interpretation relating layers to temporal processing remains somewhat speculative. More discussion of biophysiological mechanisms could help strengthen the proposed framework.

**Questions:**

1. Can you provide summary demographic and clinical details for the patient sample?

2. Can you include more specifics on the ECoG preprocessing pipeline?

3. Did you perform any optimization of encoding model parameters and architectures?

4. Have you compared model performance to any other neural language models besides GPT-2?

5. Can you elaborate on the biophysical mechanisms that could explain the layer-timing relationships observed?

**Details Of Ethics Concerns:**

The authors used brain activity data from nine humans.

---

> ### Author Response · Authors · 2023-11-22
> **Part 1 of response**
>
> We are deeply appreciative of the time and attention you have dedicated to thoroughly reviewing our paper, and for your insightful remarks. We have responded to the comments on a point-by-point basis. We sincerely apologize for our inability to upload our response at an earlier time, which would have allowed for a more timely discussion. This delay was largely due to the substantial number of reviewers (5) for our work, and the time it took us to consolidate our responses to all their comments. We highly value your understanding regarding this matter and remain committed to considering and addressing all pertinent feedback to improve our work.
>
> We would like to bring to your attention that we have made reference to several new figures that were generated as part of our response. These additional visual aids can be found in the supplementary file that we have uploaded, and you have access to this supplementary material.
>
> Q1:
> While the authors state they used 10 patients, details are lacking on the specific demographic and clinical characteristics of the patient sample. This could impact the generalizability of the findings.
>
> A1:
> We thank the reviewer for this comment and agree that not including this result obscures the generalizability of our findings. We have added a table to the supplementary material (Supp Table T1), which includes patient age, ethnicity, gender and language. We included a table in the supplementary materials with clinical characteristics of the patients and will make sure we add a reference to it in the main body of the paper (Supp Table T2).
>
> Q2:
> Preprocessing steps for the ECoG data should be described in more detail (e.g. filtering, re-referencing, artifact rejection).
>
> A2:
> We thank the reviewer for this comment. We do have a section on the preprocessing of the ECoG data in the Supplementary, which we have copied below.  Given the opportunity we will add a reference to this section in the main body of the paper.
>
> A.2 PREPROCESSING
> 66 electrodes from all patients were removed due to faulty recordings. Large spikes in the electrode signals, exceeding four quartiles above and below the median, were removed, and replacement samples were imputed using cubic interpolation. We then re-referenced the data to account for shared signals across all electrodes using the Common Average Referencing (CAR) method or an ICA based method (based on the participant’s noise profile). High-frequency broadband (HFBB) power provided evidence for a high positive correlation between local neural firing rates and high gamma activity. Broadband power was estimated using 6-cycle wavelets to compute the power of the 70- 200 Hz band (high-gamma band), excluding 60, 120, and 180 Hz line noise. Power was further smoothed with a Hamming window with a kernel size of 50 ms.
>
> Q3:
> The encoding model parameters, such as context length, smoothing windows, regularization, could be optimized more thoroughly. Ablations could be performed to test the effect of these modeling choices.
>
> A3:
> We thank the reviewer for their comment. We followed the encoding procedure of Goldstein et. al 2022, where they predict the brain from GPT2 last layer embeddings. We did this so that our results would build upon what they had observed for the last layer. We also wanted to avoid overfitting our model as we felt this might make our results less generalizable. However, we did verify robustness against different context lengths in the input to GPT2-XL (see Supplementary Figure S2)  and window sizes of neural signals centered around word onset (see Supplementary Figure S12).
>
> Goldstein, A., Zada, Z., Buchnik, E., Schain, M., Price, A., Aubrey, B., ... & Hasson, U. (2022). Shared computational principles for language processing in humans and deep language models. Nature neuroscience, 25(3), 369-380.
>
> We continue our response in the next comment.

---

> > ### Author Response · Authors · 2023-11-22
> > **Part 2 of response**
> >
> > Q4:
> > The linear interpolation analysis addresses one type of baseline model, but comparisons to other neural language models (BERT, ELMo etc) would be informative.
> >
> > A4:
> > We thank the reviewer for their comment. We acknowledge that the linear interpolation analysis measures our results against a linear baseline model. We therefore took the reviewer’s advice and ran our analysis on BERT embeddings. However, it is important to note that there are several considerations that arise when generating an embedding from BERT, considerations that are not present with GPT2.  Firstly, BERT is designed to train on complete sentences. Given that spoken language doesn't always present in full sentences, we were required to manually intervene in punctuation (aided by the nltk package). Nevertheless, this implies that the word embeddings carry information from subsequent words. From a cognitive viewpoint, this constitutes an entirely different model.
> >
> > We extracted embedding from Bert-Large’s 24  layers for predictable words and repeated the analyses. Using BERT,  the results for mSTG,  aSTG and IFG were not significant (p>0.5). Surprisingly, TP demonstrated negative correlation (r=-.6,p<.003). The plots can be seen in Supp file S1.
> >
> > Q5:
> > The theoretical interpretation relating layers to temporal processing remains somewhat speculative. More discussion of biophysiological mechanisms could help strengthen the proposed framework.
> >
> > A5:
> > We are thankful for the reviewer's comment. We completely concur that the biological plausibility of transformer-based architecture warrants discussion in our paper. Given the opportunity, we will elaborate more on this topic, reviewing work that specifically connects brain neural mechanisms (at the biophysiological level) with transformers, such as the study by Kozachkov et al. (2023), and with deep learning architectures in general, as researched by Beniaguev et al. (2023).
> > Kozachkov, L., Kastanenka, K. V., & Krotov, D. (2023). Building transformers from neurons and astrocytes. Proceedings of the National Academy of Sciences, 120(34), e2219150120
> > Beniaguev, D., Segev, I., & London, M. (2021). Single cortical neurons as deep artificial neural networks. Neuron, 109(17), 2727-2739.

---

> > > ### Comment · Reviewer_4Wkk · 2023-11-22
> > > **Final Response**
> > >
> > > Thanks to the author for the detailed response. I have no further comments to make based on your reply.

---

> > > > ### Author Response · Authors · 2023-11-23
> > > > **Replicating results for other autoregressive language models (LLaMA-7 billion parameters)**
> > > >
> > > > Relating to your question about other neural language models, just now we replicated our results for a recently published, larger language model (LLaMA 7 billion parameters) we generated embeddings from its 33 layers using the podcast words. We get significant correlations for the lag layer result for the aSTG, IFG and TP (r=0.587, r=0.825, r=0.639; p< 10e-3, p<10e-8, p<10e-4). We get a similar range of times of peak encoding correlation for these three ROIs as well (0-300ms for IFG and TP, -50ms to 100ms for aSTG). The layers all peak around 0ms in the mSTG which also reproduces our results. Thus, our results were replicated for LLaMA 7b, but not for BERT, further strengthening our claim that autoregressive deep language models are powerful cognitive models of language comprehension in the human brain. The results can be seen in Supp Fig. S13.

---

### Official Review · Reviewer_MbTk · 2023-11-01

**Soundness:** 2 fair
**Presentation:** 2 fair
**Contribution:** 2 fair
**Rating:** 3
**Confidence:** 4

**Summary:**

This paper compares recordings from intracranial ECoG electrodes while nine epilepsy patients listened to a twenty minute radio program in which a speaker tells a story to context embeddings across different layers from GPT2-XL.  A model is trained to predict high-frequency power across electrodes at moments before and after onset of each word from the contextual embedding at each of 48 layers of GPT2-XL.  A 200 ms window was slid across the four seconds adjacent to the presentation of each word; given the rate of speech this interval overlaps with an irregular number of adjacent words.  For words that were rated as predictable by GPT2-XL, the time point at which the model was most predictive changed systematically across layers by about 200 ms. Similar results were observed for other regions believed to be important in language processing, but not in mTSG.  Results for words that were not well predicted by the language model were much less consistent.

**Strengths:**

People are very interested in the connection between language models and neural signals.  The use of ECoG enables a close examination of temporal response profiles.

**Weaknesses:**

There are really important methodological details that are difficult to find or not described.  For instance, the neural signal that is being predicted is high-frequency broad band power (Appendix A2).  This should be mentioned in the main text.

There is a preselection for electrodes that is not explained beyond this statement: ``We selected electrodes that had significant encoding performance for static embeddings (GloVe) (corrected for multiple comparisons).''  This needs to be explained in much more detail.  What exactly does that mean?  How many electrodes were excluded?

In addition, the results are not readily interpretable.  It's unclear what to make of the sequential match of different layers to the same neural signal within a region.  Are we to conclude that each electrode samples from 48 functional layers within IFG?  And also within aSTG?  What could that possibly mean?

**Questions:**

How much of the variance in the neural response captured by the model can be accounted for via ERPs?   Does HFBB activity correlate with, say, N4 amplitude?  If HFBB activity was averaged *across* electrodes within region, would one still observe the same result?

How much of the variance can be accounted for by the prosody of the speaker in the radio program?   Presumably the speaker signals many things, including the predictability of words.  How variable is the timing in the rate of speech in this sample?

In ECoG studies with epilepsy patients, large effects at the population level can be driven by a small number of electrodes within one or two patients.  How consistent are the results across participants?

---

> ### Author Response · Authors · 2023-11-22
>
> We are deeply appreciative of the time and attention you have dedicated to thoroughly reviewing our paper, and for your insightful remarks. We have responded to the comments on a point-by-point basis. We sincerely apologize for our inability to upload our response at an earlier time, which would have allowed for a more timely discussion. This delay was largely due to the substantial number of reviewers (5) for our work, and the time it took us to consolidate our responses to all their comments. We highly value your understanding regarding this matter and remain committed to considering and addressing all pertinent feedback to improve our work.
>
> We would like to bring to your attention that we have made reference to several new figures that were generated as part of our response. These additional visual aids can be found in the supplementary file that we have uploaded, and you have access to this supplementary material.
>
> Reviewer Comments and Responses:
>
> Q1:
> There are really important methodological details that are difficult to find or not described. For instance, the neural signal that is being predicted is high-frequency broad band power (Appendix A2). This should be mentioned in the main text.
>
> A1:
> We thank the reviewer for this comment. Given the opportunity we will add information about the neural signal that is being predicted to the main body of the text and will add references to the supplementary for more details on neural signal preprocessing.
>
> Q2:
> There is a preselection for electrodes that is not explained beyond this statement: ``We selected electrodes that had significant encoding performance for static embeddings (GloVe) (corrected for multiple comparisons).'' This needs to be explained in much more detail. What exactly does that mean? How many electrodes were excluded?
>
> A2:
> We thank the offer for this comment. We include a more in depth description of electrode selection in the appendix but did not refer to it in the main text. Given the opportunity we will fix this so that this information is more easily accessible. We have copied the section below, with modifications in bold to add clarity. We will incorporate these modifications into the text.
>
> A.8 ELECTRODE SELECTION
> **As stated in A.1, we began with 1106 electrodes on the left hemisphere, of which 66 were removed due to faulty recordings leaving 1040 (A.2). To identify significant electrodes from this set**, we used a nonparametric statistical procedure with correction for multiple comparisons (Nichols & Holmes, 2001). At each iteration, we randomized each electrode’s signal phase by sampling from a uniform distribution. This disconnected the relationship between the words and the brain signal while preserving the autocorrelation in the signal. We then performed the encoding procedure for each electrode (for all lags). We repeated this process 5000 times. After each iteration, the encoding model’s maximal value across all lags was retained for each electrode. We then took the maximum value for each permutation across electrodes. This resulted in a distribution of 5000 values, which was used to determine the significance for all electrodes. For each electrode, a p-value was computed as the percentile of the non-permuted encoding model’s maximum value across all lags from the null distribution of 5000 maximum values. Performing a significance test using this randomization procedure evaluates the null hypothesis that there is no systematic relationship between the brain signal and the corresponding word embedding. This procedure yielded a p-value per electrode, corrected for the number of models tested across all lags within an electrode. To further correct for multiple comparisons across all electrodes, we used a false-discovery rate (FDR). Electrodes with q-values less than .01 are considered significant. This procedure identified 160 electrodes in early auditory areas, motor cortex, and language areas in the left hemisphere **(hence 880 of the 1040 were excluded for being insignificant)**. We used subsets of this list corresponding to the IFG (n=46), TP (n=6), aSTG (n=13) and mSTG (n=28).
>
> We continue our response in the following comment.

---

> > ### Author Response · Authors · 2023-11-22
> > **Part 2 of response**
> >
> > Q3:
> > In addition, the results are not readily interpretable. It's unclear what to make of the sequential match of different layers to the same neural signal within a region. Are we to conclude that each electrode samples from 48 functional layers within IFG? And also within aSTG? What could that possibly mean?
> >
> > A3:
> > We sincerely appreciate the reviewer's comment and, if given the opportunity, we plan to address it in the discussion section of our paper. This response is multifaceted. Firstly, there is an overlap in the brain regions that process different aspects of speech. For example, while the Inferior Frontal Gyrus (IFG) is typically associated with processing syntax and semantics of speech, it is also involved in the processing of phonetic information (Turker et al., 2023). As another instance, the middle Superior Temporal Gyrus (mSTG), often associated with low-level acoustic features, is also connected with syntax processing (Turker et al., 2023). Thus, while certain areas may be predisposed towards specific levels of processing (e.g., IFG towards syntactic-semantic level), they also process other aspects of language.
> > Secondly, while the embeddings induced by different layers bias towards specific linguistic features (Tenny et al., 2019), they also significantly correlate with each other. Therefore, it is not surprising that some linguistic information can be extracted from multiple layers. In conclusion, considering that the embeddings encode multiple facets of language and that different regions of interest (ROIs) overlap in the aspects of language they encode, it is plausible that different ROIs correlate with the same layers, albeit with different biases. However, the temporal dynamic we demonstrate with areas and between areas is unique.
> > Turker, S., Kuhnke, P., Eickhoff, S. B., Caspers, S., & Hartwigsen, G. (2023). Cortical, subcortical, and cerebellar contributions to language processing: A meta-analytic review of 403 neuroimaging experiments. Psychological Bulletin
> > Tenney, I., Das, D., & Pavlick, E. (2019). BERT rediscovers the classical NLP pipeline. arXiv preprint arXiv:1905.05950.
> >
> > Q4:
> > How much of the variance in the neural response captured by the model can be accounted for via ERPs? Does HFBB activity correlate with, say, N4 amplitude? If HFBB activity was averaged across electrodes within region, would one still observe the same result?
> >
> > A4:
> > In their 2022 paper, Goldstein et al. carried out ERP analysis on the HFBB of this dataset and demonstrated an N400 effect. However, because their work does not address the correlation with the different GPT2 layers, we believe it falls outside the scope of the present study. Our primary finding revolves around the correspondence between temporal dynamics in the brain and the layered structure (or sequence of embeddings) of GPT2. We demonstrate that the deeper the layer is, the later the embedding it induces correlates with the brain. This suggests a shared dynamic between the brain and GPT-2 computations or representaions. For greater clarity, we have scaled the encodings to 1 in plots 2 and 3, thereby emphasizing the importance of temporal dynamics rather than focusing on determining which layer induces the optimal correlation with the brain.
> > Following the Reviewer’s suggestion we averaged the neural signal per ROI and repeated the analysis. We replicated the effect for the IFG, TP and aSTG [(r=0.683,p<0.001), (r=0.863,p<0.001), (r=0.519, p<0.001) respectively). If given the opportunity we will report these results in the manuscript. The current plots can be seen at Supp S8.
> > Goldstein, A., Zada, Z., Buchnik, E., Schain, M., Price, A., Aubrey, B., ... & Hasson, U. (2022). Shared computational principles for language processing in humans and deep language models. Nature neuroscience, 25(3), 369-380.
> >
> > Q5:
> > How much of the variance can be accounted for by the prosody of the speaker in the radio program? Presumably the speaker signals many things, including the predictability of words. How variable is the timing in the rate of speech in this sample?
> >
> > A5:
> > We appreciate the reviewer’s question. In fact, it aligns with a research study we are currently conducting, where we examine models that also capture low-level acoustic features associated with prosody. The idea of exploring the association between acoustic features and predictability is indeed intriguing. However, as mentioned in our previous responses, the primary focus of this manuscript is the correlation between the temporal dynamics in the brain and the hierarchy of the GPT-2 layers. Consequently, we believe that these points fall outside the purview of this particular paper.
> >
> > We continue our response in the following comment.

---

> > > ### Author Response · Authors · 2023-11-22
> > > **Part 3 of response**
> > >
> > > Q6:
> > > In ECoG studies with epilepsy patients, large effects at the population level can be driven by a small number of electrodes within one or two patients. How consistent are the results across participants?
> > >
> > > A6:
> > > We appreciate the reviewer's comment and suggestions. Following the reviewer’s question, we repeated the analyses per-patient. We selected patients and ROIs with at least 5 electrodes. For this reason, we do not include the TP, which has 4 electrodes from 1 patient and the remaining 2 from 2 others. For the most part our results are replicable at the individual patient level. Overall, the IFG shows a larger time scale than the aSTG and the aSTG shows a larger time scale than the mSTG. We still get high correlations for IFG and aSTG. Moreover, the shapes of the encoding plots for mSTG are very different from those for IFG and aSTG with the former having a very short time scale compared to the latter two. Given the opportunity we will report the results in the paper (see Supp S9-11).

---

> > > > ### Comment · Reviewer_MbTk · 2023-11-23
> > > > **response**
> > > >
> > > > I have read the response of the authors and thank them for their clarification.
> > > >
> > > > I remain unconvinced that this paper shows a deep correspondence between GPT-XL and ECoG signals in the human brain.
> > > > The methods are *really* complicated and the ECoG signal does not have outstanding spatial resolution, plus there's a lot of heterogeneity across patients, for unavoidable reasons.  There's either a simple, neurophysiological explanation for this or it's a type I error.  Given the number of choices that need to be made to arrive at a method this complicated, I am seriously worried about a type I error.  I don't find the responses compelling.  The response to Q3 dodges the central critique---how do 48 layers meaningfully fit within a brain region (measured with electrodes that are several millimeters).  The simple answer is that there is no meaningful correspondence. The response to Q4 also dodges the question.  Why couldn't the results be understandable as a slightly later N4 at different regions, where the sources of variability underlying the N4 would be picked up at different layers of the model?  This would be a superior contribution to the current work in that it would at least point to a specific circuit-level mechanism.  Personally, my guess is that the speaker is conveying subtle aspects of meaning via prosody, and that's the sole reason there is an effect (as the authors point out mSTG and IFG are involved in phonetic processing).  If there were any kinds of control (at least more than one radio show!) we could feel more confident in this result.

---

> > > > > ### Author Response · Authors · 2023-11-23
> > > > >
> > > > > We thank the reviewer for their response. We aimed for simplicity in our procedures as we will highlight now. First the preprocessing was not tailored for this manuscript. The preprocessing procedure as well as the embedding extraction procedure were adopted from Goldstein et al. (2022) (exactly to avoid any concern over "overfitting" the preprocessing to the effect seen in our manuscript). Second, the encoding procedure (which was also adopted from the procedure in Goldstein et al. 2022) is based on simple linear regression. We chose this procedure as it is the simplest way to map embeddings to the brain that we are aware of. The contribution of this work goes above and beyond previous papers (like Goldstein et al.) as we show a similarity in the temporal dynamic in the brain and the different layers of GPT2-XL.
> > > > >
> > > > > We introduce several control analyses in the original manuscript as well as new controls thanks to reviewers' comments. For example, we did analyses to show our results are robust to hyperparameters like the context length for embedding generation (Supp S2) and smoothing window size applied to the neural signal (Supp S12). Moreover, during the current revision we replicated our results using embeddings from a different (larger) autoregressive language model (LLAMA - Supp S13) However, the results are not replicated for a non-autoregressive language model (BERT, S1) nor for the linearly interpolated embeddings (Supplementary Figure 9 in the original manuscript). This challenges the claim that there is "either a simple, neurophysiological explanation for this or it's a type I error" or that there is a prosody based explanation. If our results had one of these simple explanations, we would expect to replicate our results for an arbitrary choice of language model embeddings.
> > > > >
> > > > > The novelty of our results lies in the alignment between the layers and the temporal signal within the brain. Autoregressive Language models represent a new way of studying neural language processing, and the task of understanding the information encoded within different layers, or the interpretability of deep learning models, is an active field of research. Our work implies that applying these models to neural data as we have in this paper can help to better understand the human language comprehension process, while also improving our understanding of deep language models. Following the reviewer’s comments, and if given the opportunity, we will address it in the discussion section of the paper.

---

### Official Review · Reviewer_4TH4 · 2023-11-01

**Soundness:** 4 excellent
**Presentation:** 4 excellent
**Contribution:** 3 good
**Rating:** 8
**Confidence:** 4

**Summary:**

With the emergence of LLMs the previously popular idea of comparing visual processing in DNNs with human ventral stream has jumped onto the next logical opportunity: comparing progression of representations in LLM with brain responses. However, as it was discovered, spatially the alignment is not that clear, it's not like you can track which brain area corresponds to which "layer" of an LLM.

In this paper the authors argue that the desired alignment can be tracked, but through time, not through space. They use ECoG recordings that provide them with the ability to track neural signal through time, while the previous studies, those that claimed there is no great alignment, could not do that because they were using fMRI recordings that have very poor temporal resolution in comparison to ECoG.

In the rest of the document authors show how the best-correlating layers seem to show a distinctive pattern: early LM layers best predict brain activity at early times, mid LM layers - brain activity at mid times and late LM layers - brain activity at later times. This very much seems to confirm the hypothesis that as the representation of a word evolves thought the layers of a DLM, it matches to representations found in the brain in the later timestamps, perhaps because a word evolves in some regards similarly in out brain.

This highlights an important funding that differently from visual processing, language computation in the brain is temporally hierarchical and is localised in the same area, bringing forward the potential important of recurrent computation for this function.

**Strengths:**

This paper presents a strong case for the alignment between temporal evolution of a representation of a word and its evolution through the layers of a DLM. The work is well-structures, presents the material clearly and logically.

In my opinion this is a clear and good contribution to the field of NeuroAI and clearly presents a finding that highlight some truths about how the brain works. I will try to poke some holes in the author's argument in the questions below, but overall I think this result is solid and I find it to be of interest and significance.

**Weaknesses:**

(1) Maybe I've missed it, but did you measure the decoding accuracy (Decoder : ECoG -> Words). Before we we ask if we can reconstruct those representations it would be nice to know that they actually carry information about the words and are not just unrelated neural processes. Without this, from Reconstruction : Embeddings -> Signal we know that we can reconstruct and capture the temporal dynamics in the brain signal, but this does not necessarily mean that this dynamics is relevant to language processing (I understand that the area where the signal is from is a language area, but who knows what else it might be doing that is contributing to the signal we are so diligently are trying to reconstruct).

(2) What happens with not-so-much-predictable words? Why?

**Questions:**

(1) How well would you be able to classify (using as deep and powerful model as needed) individual words from ECoG signal alone? In my mind (in)ability to do this would be informative in terms of whether the recorded activity actually contains strong enough signal to claim that it carries any bits of language representation in the brain.

(2) What is the canonic path of a word through the brain? Which areas are being activated in which order according to the modern day knowledge on this? The reader would benefit of a figure explaining this, same as we've seen in all the vision papers.

(3) Are there differences between spoken words and written words? Do their brain-paths converge at some point? Where? Do we believe that representations are different depending on whether a word was perceived auditorily and visually? If semantic representation is what we are after, should we only compare representations "after" this point of convergence once the word took its abstract semantic representation form that is divorced the "mechanics" of delivery?

(4) How did you handle multiple subjects: were their data put into one large pot, or each subject was treated separately (by training subject-specific models)?

(5) Figure 2: How would the whole lag-layer-correlation picture look like if you would shuffle the words that are being decoded? I am trying to understand if this picture we are seeing is indeed due to language structure and representational similarities between brains and DLMs, and whether this nice picture would disappear if we carefully and deliberately remove the sought signal from the data?

(6) Figure 2: Also, if we, once again, remove the signal from the data via some sort of permutation test, how all these plots will look like? Some other shape - which one? Just flat - why? The answer to this question will, in turn, lead to a question about why those shapes look like they do in the absence of the true signal.

(7) Figure 2: And one more here: how high the correlation would rise for the data we know should have no correlation? Would it be actually 0 all the way from -4000 to 4000, or because of some general dynamics and the way how to the linear readout models are trained some positive correlation will still be observed? How high?

(8) Could the correlation be explained by, say, non-sparsity of the representation in DLM in the mid layers, or some other technical reason, and not by the actual match between representations?

---

> ### Author Response · Authors · 2023-11-22
>
> We are deeply appreciative of the time and attention you have dedicated to thoroughly reviewing our paper, and for your insightful remarks. We have responded to the comments on a point-by-point basis. We sincerely apologize for our inability to upload our response at an earlier time, which would have allowed for a more timely discussion. This delay was largely due to the substantial number of reviewers (5) for our work, and the time it took us to consolidate our responses to all their comments. We highly value your understanding regarding this matter and remain committed to considering and addressing all pertinent feedback to improve our work.
>
> We would like to bring to your attention that we have made reference to several new figures that were generated as part of our response. These additional visual aids can be found in the supplementary file that we have uploaded, and you have access to this supplementary material.
>
> Reviewer Comments and Responses:
>
> Q1: Maybe I've missed it, but did you measure the decoding accuracy (Decoder : ECoG -> Words). Before we we ask if we can reconstruct those representations it would be nice to know that they actually carry information about the words and are not just unrelated neural processes. Without this, from Reconstruction : Embeddings -> Signal we know that we can reconstruct and capture the temporal dynamics in the brain signal, but this does not necessarily mean that this dynamics is relevant to language processing (I understand that the area where the signal is from is a language area, but who knows what else it might be doing that is contributing to the signal we are so diligently are trying to reconstruct).
>
> A1:
> We appreciate the reviewer's inquiry. Indeed, Goldstein et al. (2022) undertook decoding on this data set, demonstrating the feasibility of classifying the identity of words from the neural signal. In fact, they revealed that the contextual embedding induced by the final layer could be predicted from the neural signal and by comparing the predicted embedding it is possible to classify the identity of the words. Therefore, the reconstruction of the embedding indeed holds word-specific information. Given the opportunity we will cite these results and explain their importance in the interpretation of our results.
>
> Goldstein, A., Zada, Z., Buchnik, E., Schain, M., Price, A., Aubrey, B., ... & Hasson, U. (2022). Shared computational principles for language processing in humans and deep language models. Nature neuroscience, 25(3), 369-380.
>
> Q2:
> What happens with not-so-much-predictable words? Why?
>
> A2:
> We thank the reviewer for this question. We realized that we did not refer to this result in the manuscript. These analyses  can be found in Supp S4&S5).If given the opportunity we will refer to it in the revision. Interestingly, the temporal ordering of the layers is robust to the choice of predictable or unpredictable words, however it occurs later for unpredictable words (between 0 and 250ms for predictable and between 250ms and 500ms for unpredictable). Though we have not explored this effect in depth, it may indicate a neural response that occurs in response to prediction error.
>
> Q3:
> How well would you be able to classify (using as deep and powerful model as needed) individual words from ECoG signal alone? In my mind (in)ability to do this would be informative in terms of whether the recorded activity actually contains strong enough signal to claim that it carries any bits of language representation in the brain.
>
> A3:
> We thank the reviewer for the question.  We believe our answer to A1 applies here as well but please let us know if there is something we missed and we will happily address it.
>
> Q4:
> What is the canonic path of a word through the brain? Which areas are being activated in which order according to the modern day knowledge on this? The reader would benefit of a figure explaining this, same as we've seen in all the vision papers.
>
> A4:
> To the best of our knowledge, a definitive neural architecture for word processing does not currently exist. Furthermore, contextual embeddings are not limited solely to representing the identity of a word, but also consider its meaning in relation to its context. However, in a study conducted by Hasson, Chen, and Honey (2015), they propose a model of language comprehension and provide a diagram in Figure 2B (right) that illustrates the flow of information. Interestingly, this diagram demonstrates a similar pathway to the one we describe in our paper (mSTG->aSTG->TP, IFG). Additionally, further supporting our claims is the observation that the order of regions of interest (ROIs) in this pathway correlates with the temporal receptive field, which we measure in our manuscript through the spread of the peaks.
> Hasson, U., Chen, J., & Honey, C. J. (2015). Hierarchical process memory: memory as an integral component of information processing. Trends in cognitive sciences, 19(6), 304-313.
>
>
> We continue in the next comment.

---

> ### Author Response · Authors · 2023-11-22
>
> Q5:
> Are there differences between spoken words and written words? Do their brain-paths converge at some point? Where? Do we believe that representations are different depending on whether a word was perceived auditorily and visually? If semantic representation is what we are after, should we only compare representations "after" this point of convergence once the word took its abstract semantic representation form that is divorced the "mechanics" of delivery?
>
> A5:
> A thorough literature review on the relationship between comprehension of written and spoken language is beyond the scope of our paper, which specifically focuses on speech comprehension. However, Regev, Honey, Simony, & Hasson (2013) did conduct a study comparing the brain dynamics of written and spoken language comprehension. Their findings revealed that certain areas, such as the IFG, were consistently activated regardless of the medium of language. In contrast, other areas, such as the STG, were specifically activated for auditory language, and the visual cortex exhibited unique activation for written words. In alignment with these findings, we commence our paper by emphasizing the significance of the IFG, as it remains invariant across different language modalities and later look at areas that are specific for auditory comprehension (such as STG).
> Regev, M., Honey, C. J., Simony, E., & Hasson, U. (2013). Selective and invariant neural responses to spoken and written narratives. Journal of Neuroscience, 33(40), 15978-15988.
>
> Q6:
> How did you handle multiple subjects: were their data put into one large pot, or each subject was treated separately (by training subject-specific models)?
>
> A6:
> We thank you for this question and the opportunity to clarify this point in our work. We trained a separate encoding model for each electrode. Thus we do not need to explicitly address the different patients.
>
> Q7:
> Figure 2: How would the whole lag-layer-correlation picture look like if you would shuffle the words that are being decoded? I am trying to understand if this picture we are seeing is indeed due to language structure and representational similarities between brains and DLMs, and whether this nice picture would disappear if we carefully and deliberately remove the sought signal from the data?
>
> A7:
> We thank the reviewer for their insight. Following the comment we shuffled the words, and by doing so mismatching the words from their neural signal and repeated the analyses. None of the ROI yielded significant correlation between the lag that yields the maximal correlation and the layer index. The plots could be seen at Supp S6. Given the opportunity we will include this new analysis in the paper.
>
> Q8:
> Figure 2: Also, if we, once again, remove the signal from the data via some sort of permutation test, how all these plots will look like? Some other shape - which one? Just flat - why? The answer to this question will, in turn, lead to a question about why those shapes look like they do in the absence of the true signal. Figure 2: And one more here: how high the correlation would rise for the data we know should have no correlation? Would it be actually 0 all the way from -4000 to 4000, or because of some general dynamics and the way how to the linear readout models are trained some positive correlation will still be observed? How high?
>
> A8:
> We thank the reviewer for these questions. We performed an analysis where we phase shuffled the neural signal and re-ran our encoding procedure. The lag layer correlation effect disappears. We also see maximum correlations in the coding plots < 0.05 (much less than for true signal) and none of them turned significant (p>0.1).  The plots can be seen at Supp S7. Given the opportunity we will include this new analysis in the paper.
>
> Q9:
> Could the correlation be explained by, say, non-sparsity of the representation in DLM in the mid layers, or some other technical reason, and not by the actual match between representations?
>
> A9:
> We thank the reviewer for this comment. It is always the case that there are confounders we could not control (because we did not think about them). However, the control analysis we introduce in the original paper, as well as the controls suggested by the current reviewers give us confidence in the results.

---

### Official Review · Reviewer_r5Ne · 2023-11-01

**Soundness:** 3 good
**Presentation:** 3 good
**Contribution:** 3 good
**Rating:** 6
**Confidence:** 5

**Summary:**

There is a fastly growing literature analyzing how deep language models' representations have predictive power over fMRI brain measurements. These papers typically train a regressor model (typically a linear regressor) to predict the fMRI measurements from the neural language models' representations. Current linguistic encoding models on analyzing deep language model representations have shown that these models learn rich linguistic knowledge within their representations. The work done in neuroscience has shown alignment between language models with layer hierarchy and high-level language brain regions for both fMRI and MEG recordings. They then evaluate predictive power by measuring the correlation between predictions and actual measurements.

This paper contributes to that literature by showing evidence that the layered hierarchy of deep language models (DLM) may be used to model the temporal dynamics of language comprehension in the brain with the help of electrocorticography (ECoG) recordings. Further, the authors demonstrate a strong correlation between DLM layer depth and the time at which it is most predictive of the human brain.

**Strengths:**

The paper contains the following key contributions:

* Provide evidence that the layered hierarchy of GPT2-XL can be used to model the temporal hierarchy of language comprehension in high-order human language areas.
* Like earlier studies, intermediate layers are well aligned with the brain, even in the case of ECoG brain recordings.
* It also highlights some difference between the brain and transformer model, in that the brain likely relies more on recurrent processing as it does not have space to hold all past word tokens.

Originality: The idea of how contextualized GPT-2 XL model representations are aligned in the Brain and demonstrated the temporal dynamics of the hierarchy of language compression in the Brain with the help of ECoG recordings.

Quality: The paper supports its claims with enough details. The paper is well-written and easy to follow. However, all the Figures are hard to follow based on captions.

Clarity: The paper is written well. The information provided in the submission is sufficient to reproduce the results.

Significance: The idea of using GPT-2 XL model representations to investigate the temporal dynamics of language comprehension using ECoG recordings is interesting.

**Weaknesses:**

* Current work focuses more on layer-wise transformations learned by GPT2-XL map onto the temporal sequence of transformations of natural language. However, the current paper lacks in providing fine-grained details as follows: (i) Why are intermediate layers well aligned with the Brain? (ii) It could be interesting if authors could report more fine-grained details like the following study: Subba Reddy Oota, Mariya Toneva. Joint processing of linguistic properties in brains and language models, NeurIPS-2023, https://arxiv.org/pdf/2212.08094.pdf (iii) Supraword meaning analysis and layer-wise hierarchy details.

* Caucheteux et al. 2022 "Brains and algorithms partially converge in natural language processing, shown that both MLM and CLM models yield best alignment with fMRI & MEG in middle layers and reported better performance at predicting next word -> better prediction of fMRI & MEG. How does the current work differ in terms of new findings except for the use of ECoG recordings?

* It could be interesting if authors could report syntax and low-level surface feature analysis across language regions.

**Questions:**

* Did authors try the encoding with different context lengths?
* Did authors try the encoding performance with previous word representations (i.e. using w{t-1} to predict w_{t} ECoG data, similarly w_{t-2}, w_{t-3} ..)?
* Did authors try the encoding performance with low-level features, including the number of letters, phonemes, word length, word frequency, etc,. to syntax-level features?
* The authors state that they "are willing to provide our data to those interested in reproducing our experiments". Why not make the data entirely available for future work to build on these results? Is that related to some required privacy clause in the data collection?

---

> ### Author Response · Authors · 2023-11-22
> **First part of response**
>
> We are deeply appreciative of the time and attention you have dedicated to thoroughly reviewing our paper, and for your insightful remarks. We have responded to the comments on a point-by-point basis. We sincerely apologize for our inability to upload our response at an earlier time, which would have allowed for a more timely discussion. This delay was largely due to the substantial number of reviewers (5) for our work, and the time it took us to consolidate our responses to all their comments. We highly value your understanding regarding this matter and remain committed to considering and addressing all pertinent feedback to improve our work.
>
> We would like to bring to your attention that we have made reference to several new figures that were generated as part of our response. These additional visual aids can be found in the supplementary material section of openreview
>
> Reviewer Comments and Responses:
>
> Q1:
> Current work focuses more on layer-wise transformations learned by GPT2-XL map onto the temporal sequence of transformations of natural language. However, the current paper lacks in providing fine-grained details as follows: (i) Why are intermediate layers well aligned with the Brain?(ii) It could be interesting if authors could report more fine-grained details like the following study: Subba Reddy Oota, Mariya Toneva. Joint processing of linguistic properties in brains and language models, NeurIPS-2023, https://arxiv.org/pdf/2212.08094.pdf (iii) Supraword meaning analysis and layer-wise hierarchy details.
>
> A1:
> We thank the reviewer for this comment. Previous work noted  the higher alignment between intermediate layers and the brain during language processing (Toneva and Wehbe 2019 and Caucheteux and King 2022, for example). We also replicated this result as can be shown in Figures 2 and 5, however, the focus of our paper is different - our primary finding revolves around the correspondence between temporal dynamics in the brain and the layered structure (or sequence of embeddings) of GPT2. We demonstrate that the deeper the layer is, the later the embedding it induces correlates with the brain. This suggests a shared dynamic between the brain and GPT-2 computations or representaions. For greater clarity, we have scaled the encodings to 1 in plots 2 and 3, thereby emphasizing the importance of temporal dynamics rather than focusing on determining which layer induces the optimal correlation with the brain.
>
> Caucheteux, C., King, JR. Brains and algorithms partially converge in natural language processing. Commun Biol 5, 134 (2022). https://doi.org/10.1038/s42003-022-03036-1
> Toneva, Mariya, Leila, Wehbe. "Interpreting and improving natural-language processing (in machines) with natural language-processing (in the brain)." Advances in Neural Information Processing Systems. 2019.
>
> Q2:
> Caucheteux et al. 2022 "Brains and algorithms partially converge in natural language processing, shown that both MLM and CLM models yield best alignment with fMRI & MEG in middle layers and reported better performance at predicting next word -> better prediction of fMRI & MEG. How does the current work differ in terms of new findings except for the use of ECoG recordings?
>
> A2:
> We thank the reviewer for the question. Indeed, Caucheteux et al. (2022), also demonstrates correlation with embedding induced by deep language models and the brain. However, here we focus on the shared dynamic between the brain and the flow of information through the GPT-2 layers (see A1 for more details).
>
> Q3:
> It could be interesting if authors could report syntax and low-level surface feature analysis across language regions.
>
> A3:
> We fully acknowledge the intriguing nature of assessing the predictive ability of LLM-induced embeddings in comparison to symbolic language representations such as syntax and part-of-speech (PoS). In fact, we are actively engaged in the writing of a manuscript focused specifically on this topic. However, in the context of the present manuscript, we believe that diverting the discussion towards a comparison with symbolic representations would deviate from its primary scope and purpose.
>
> Q4:
> Did authors try the encoding with different context lengths?
>
> A4:
> We thank the reviewer for this question. In response to this request we generated embeddings by running GPT2-XL on the podcast data with input sizes of 500 tokens and 100 tokens. We then re-ran our analysis for correct words in the inferior frontal gyrus (IFG). The global trends are robust to different context lengths during embedding generation (see in Supp S2). If given the opportunity we will report this analysis in the manuscript.
>
> We continue our response in the following comment.

---

> > ### Author Response · Authors · 2023-11-22
> > **Part 2 of response**
> >
> > Q5:
> > Did authors try the encoding performance with previous word representations (i.e. using w{t-1} to predict w_{t} ECoG data, similarly w_{t-2}, w_{t-3} ..)?
> >
> > A5:
> > We thank the reviewer for this question. In our paper, we followed Goldstein et al. (2022) and chose the representation of the w_{t-1} word. Following the reviewer question we repeated the main analysis for embeddings extracted for w{t} w{t-2}, w{t-3}. The results can be seen at Supp S3. If given the opportunity we will include them in the paper.
> > Goldstein, A., Zada, Z., Buchnik, E., Schain, M., Price, A., Aubrey, B., ... & Hasson, U. (2022). Shared computational principles for language processing in humans and deep language models. Nature neuroscience, 25(3), 369-380.
> >
> > Q6:
> > Did authors try the encoding performance with low-level features, including the number of letters, phonemes, word length, word frequency, etc,. to syntax-level features?
> >
> > A6:
> > We fully acknowledge the intriguing nature of assessing the predictive ability of LLM-induced embeddings in comparison to symbolic language representations (e.g., number of letters, phonemes, word length, word frequency, Part of Speech etc.). In fact, we are actively engaged in the writing of a manuscript focused specifically on this topic. However, in the context of the present manuscript, we believe that diverting the discussion towards a comparison with symbolic representations would deviate from its primary scope and purpose. Our primary finding revolves around the correspondence between temporal dynamics in the brain and the layered structure (or sequence of embeddings) of GPT2. We demonstrate that the deeper the layer is, the later the embedding it induces correlates with the brain. This suggests a shared dynamic between the brain and GPT-2 computations or representaions.  For greater clarity, we have scaled the encodings to 1 in plots 2 and 3, thereby emphasizing the importance of temporal dynamics rather than focusing on determining which layer induces the optimal correlation with the brain.
> >
> > Q7:
> > The authors state that they "are willing to provide our data to those interested in reproducing our experiments". Why not make the data entirely available for future work to build on these results? Is that related to some required privacy clause in the data collection?
> >
> > A7:
> > We sincerely value the reviewer's comment. Given the sensitive nature of the clinical data collected, the institutions responsible for the data collection have opted to share the data through collaborative efforts. Adhering to this policy, they have so far shared the data with numerous institutions.

---

### Official Review · Reviewer_rNmu · 2023-11-01

**Soundness:** 3 good
**Presentation:** 4 excellent
**Contribution:** 3 good
**Rating:** 8
**Confidence:** 4

**Summary:**

In this paper, the authors explore the relationship between the hierarchical structure of Deep Language Models (DLMs) like GPT2-XL and the temporal processing of language in the human brain. The authors use electrocorticography (ECoG) to record neural activity from participants listening to a narrative, while also feeding the same narrative to GPT2-XL. Contextual embeddings from the different layers of the DLM are extracted and used to predict neural activity in the brain. The experiment results suggest that the sequential layer processing in DLMs mirrors the timing of neural activity in human brain language areas.

**Strengths:**

The paper is written clearly and coherently, and it is well-structured with a logical chain of thought, making it easy to follow.
The
Extensive experiments show both temporal and spatial alignment between the DLMs and the brain, further enhancing the reliability of the results.
The finding that the layered hierarchy of DLMs can model the temporal hierarchy of language is innovative and could be a significant contribution to the field of developing brain-inspired LLMs with better alignment.

**Weaknesses:**

Lack of comparative baselines: The paper only uses GPT2-XL, and other language models such as BERT, as well as recently released LLMs (e.g., LLaMA, Vicuna) should be considered for comparison.

**Questions:**

I do not have additional questions at this stage

---

> ### Author Response · Authors · 2023-11-22
>
> We are deeply appreciative of the time and attention you have dedicated to thoroughly reviewing our paper, and for your insightful remarks. We sincerely apologize for our inability to upload our response at an earlier time, which would have allowed for a more timely discussion. This delay was largely due to the substantial number of reviewers (5) of our work, and the time it took us to run new analyses and to consolidate our responses to all their comments. We highly value your understanding regarding this matter and remain committed to considering and addressing all pertinent feedback to improve our work.
>
> We would like to bring to your attention that we have made reference to several new figures that were generated as part of our response. These additional visual aids can be found in the supplementary material section of openreview.
>
> Reviewer Comments and Responses:
>
> Q1:
> Lack of comparative baselines: The paper only uses GPT2-XL, and other language models such as BERT, as well as recently released LLMs (e.g., LLaMA, Vicuna) should be considered for comparison.
>
> A1:
> We appreciate the comment offered by the reviewer. Three foundational studies establish a connection between language comprehension in the brain and GPT-2 (Goldstein et. al 2022, Caucheteux et al.  2022, Schrimpf et al 2021). In particular, two of these papers (Goldstein et. al 2022, Schrimpf et al 2021) explicitly discuss the selection of autoregressive language models (such as GPT-2) as particularly suitable for modeling brain responses to natural language stimuli as the brain is known to make predictions about subsequent words before they are spoken On the contrary, the BERT model is bidirectional and less suitable for modeling the cognitive system, which does not have access to the future words at prediction time. However, LLAMA is autoregressive, fitting the required model criteria. We are currently generating word embeddings from LLAMA and will re-run our analysis. This process takes some time as we need to get familiar with a new tokenizer and other technical details. If given the opportunity we will add the LLAMA based results. In addition, we will include the rationale we have outlined here in the manuscript for choosing GPT-2 for modeling the process of language comprehension in the brain.
> Following the reviewer's request, we also performed an analysis for BERT embeddings. However, it is important to note that there are several considerations that arise when generating an embedding from BERT, considerations that are not present with GPT2.  Firstly, BERT is designed to train on complete sentences. Given that spoken language doesn't always present in full sentences, we were required to manually intervene in punctuation (aided by the nltk package). Nevertheless, this implies that the word embeddings carry information from future words. From a cognitive viewpoint, this constitutes an entirely different model.
> We extracted embeddings from BERT-Large’s 24  layers for predictable words. The mSTG, IFG and aSTG did not have significant results (p>0.5, p> 0.1, p>0.7). Neither the IFG nor the aSTG show the effect of increasing peak encoding correlation time with the layer index that we see with GPT2-XL embeddings. In the TP we get a significant effect (p<0.03) of a negative correlation between layer index and lag, which is unexpected. As pointed out before, there are multiple aspects that need to be considered when using BERT to model neural signal. Thus a further investigation is needed to understand these results. The BERT based plots can be seen in Supp table S1.
>
> Goldstein, A., Zada, Z., Buchnik, E., Schain, M., Price, A., Aubrey, B., ... & Hasson, U. (2022). Shared computational principles for language processing in humans and deep language models. Nature neuroscience, 25(3), 369-380.

---

> ### Author Response · Authors · 2023-11-22
> **UPDATE: LLaMA 7b results**
>
> We thank the reviewer for their suggestion to repeat our analyses using embeddings from different language models. We generated embeddings from the 33 layers of LLaMA 7b using the podcast words. The results can be seen in Supp Fig. S13. We get significant correlations for the lag layer result for the aSTG, IFG and TP (r=0.587, r=0.825, r=0.639; p< 10e-3, p<10e-8, p<10e-4). We get a similar range of times of peak encoding correlation for these three ROIs as well (0-300ms for IFG and TP, -50ms to 100ms for aSTG). The layers all peak around 0ms in the mSTG which also reproduces our results. Thus, our results were replicated for LLaMA 7b, but not for BERT, further strengthening our claim that autoregressive deep language models are powerful cognitive models of language comprehension in the human brain.

---

### Meta-Review · Area_Chair_HVx9 · 2023-12-09

**Metareview:**

The submission presents a correlation analysis of electrocorticography (ECoG) data recorded from human participants and internal representations from several deep language models, finding positive results (high correlation). An expert reviewer notes that the flexibility allotted to the study's methodology (including the complicated model architecture and the ad hoc preprocessing pipeline) increases the likelihood of finding a correlation, including a spurious one (a type-I error), though this criticism is also borne by the prior work that introduced this pipeline (Goldstein et al., 2022).

Given that the submission makes use of the prior study's methodology, the measure of its contribution is in any new insights drawn from the new comparison. In this respect, the correlational findings do not clearly make a significant contribution in providing actionable insights for understanding human speech processing. In particular, what does being able to fit a predictive model of ECoG, even a temporally consistent one, tell us that we didn't already know about the mechanisms underlying human speech processing?

**Justification For Why Not Higher Score:**

lack of interpretable conclusions about human speech processing from this correlational analysis

**Justification For Why Not Lower Score:**

N/A

---

### Decision · Program_Chairs · 2024-01-16

Reject